# Impact of ocean acidification on Arctic phytoplankton blooms and dimethylsulfide concentration under simulated ice-free and under-ice conditions

**Hussherr, Rachel[1]; Levasseur, Maurice[1]; Lizotte, Martine[1]; Tremblay, Jean-Éric[1]; Mol, Jacoba[2]; Thomas, Helmuth[2]; Gosselin, Michel[3]; Starr, Michel[4]; Miller, Lisa A.[5]; Jarniková, Tereza[6]; Schuback, Nina[6]; Mucci, Alfonso[7]**

[1] Québec-Océan and Takuvik joint UL-CNRS laboratory, Département de biologie, Université Laval, Québec, Québec G1V 0A6, Canada

[2] Department of Oceanography, Dalhousie University, Halifax, Nova Scotia B3H 4R2, Canada

[3] Institut des sciences de la mer de Rimouski, Université du Québec à Rimouski, Rimouski, Québec G5L 3A1, Canada

[4] Maurice Lamontagne Institute, Fisheries and Oceans Canada, Mont-Joli, Québec G5H 3Z4, Canada

[5] Institute of Ocean Sciences, Fisheries and Oceans Canada, Sidney, British Columbia V8L 4B2, Canada

[6] Department of Earth, Ocean and Atmospheric Sciences, University of British Columbia, Vancouver, British Columbia V6T 1Z4, Canada

[7] GEOTOP and Department of Earth and Planetary Sciences, McGill University, Montréal,
Québec H3A 0E8, Canada

*Correspondence to*: Rachel Hussherr (rachel.hussherr@gmail.com)

**Abstract.** In an experimental assessment of the potential impact of Arctic Ocean acidification on seasonal phytoplankton blooms and associated dimethylsulfide (DMS) dynamics, we incubated water from Baffin Bay under conditions representing an acidified Arctic Ocean. Using two light regimes simulating under-ice/ subsurface chlorophyll maxima (low light; Low PAR and no UVB) and ice-free (high light; High PAR + UVA + UVB) conditions, water collected at 38 m was exposed over 9 days to 6 levels of decreasing pH from 8.1 to 7.2. A phytoplankton bloom dominated by the centric diatoms *Chaetoceros* spp. reaching up to 7.5 $\mu$g chlorophyll *a* L$^{-1}$ took place in all experimental bags. Total dimethylsulfoniopropionate (DMSP$_T$) and DMS concentrations reached 155 nmol L$^{-1}$ and 19 nmol L$^{-1}$, respectively. The sharp increase in DMSP$_T$ and DMS concentrations coincided with the exhaustion of NO$_3^-$ in most microcosms, suggesting that nutrient stress stimulated DMS(P) synthesis by the diatom community. Under both light regimes, chlorophyll *a* and DMS concentrations decreased linearly with increasing proton concentration at all pH tested. Concentrations of DMSP$_T$ also decreased but only under high light and over a smaller pH range (from 8.1 to 7.6). In contrast to nanophytoplankton (2-20 $\mu$m), picophytoplankton ($\leq 2 \mu$m) was stimulated by the decreasing pH. We furthermore observed no significant difference between the two light regimes tested in term of chlorophyll *a*, phytoplankton abundance/ taxonomy, and DMSP/ DMS net concentrations. These results show that OA could significantly decrease the algal biomass and inhibit DMS production during the seasonal phytoplankton bloom in the Arctic, with possible consequences for the regional climate.

# 1 Introduction

As a result of anthropogenic emissions of carbon dioxide ($CO_2$) to the atmosphere, important transformations are observed in the global ocean, including a rise in water temperature, a decrease in ocean pH, modifications of water circulation patterns and nutrient distributions, and a loss of sea-ice in the Arctic (ACIA, 2005; Fabry et al., 2009; Macdonald et al., 2015). Due to various feedback processes, the atmospheric temperature in the Arctic region above 64°N has

warmed by 1.9°C between 1981 and 2012, a rate three times higher than the global average (ACIA, 2005; Ford et al., 2015). This phenomenon, known as the Arctic amplification (Cohen et al., 2014), is leading to the greatest regional transformations observed in the recent decades (ACIA, 2005). Given that the reduction in the extent and thickness of the sea ice cover and the acidification of surface waters are two factors that can potentially affect Arctic primary

productivity, it is important to consider the associated effects on the production of biogenic climate-active gases such as dimethysulfide (DMS).

Since the pre-industrial era, the ocean has absorbed more than one quarter of the anthropogenic $CO_2$ emitted to the atmosphere (Feely et al., 2009, and references therein). This phenomenon alters the chemistry of seawater and results in a decrease in pH. The dissolution of anthropogenic

$CO_2$ has already led to an estimated 0.1 unit decrease of pH in the global ocean surface waters (Feely et al., 2009). An additional decrease of 0.2 pH unit is anticipated by the end of the century and as much as 0.8 by 2300, depending on future net $CO_2$ emissions (Caldeira and Wickett, 2003; Raven et al., 2005; Doney et al., 2009; Feely et al., 2009). Owing to the lower buffer capacity and greater solubility of $CO_2$ in cold waters and the dilution of total alkalinity (TA) by freshwater due

to sea-ice melt and river run-off, a more rapid decline of 0.45 unit is predicted by 2100 for Arctic waters under IPCC SRES emission scenario A2 (Steinacher et al., 2009).

During the last two decades, many studies have attempted to quantify the sensitivity of marine ecosystems to ocean acidification (OA). Since the dissolution of atmospheric $CO_2$ increases its concentration in seawater, low pH-high $pCO_2$ could enhance phytoplankton growth rate by

facilitating $CO_2$ uptake and reducing the energy cost of the Carbon Concentrating Mechanism (CCM, cellular processes used by algae to overcome $CO_2$ limitation in water) in some phytoplankton species (Gao and Campbell, 2014 and references therein). Accordingly, a decrease in pH has been reported to stimulate carbon fixation by phytoplankton (Doney et al., 2009; Gao

and Campbell, 2014; Wu et al., 2014; Mackey et al., 2015; Thoisen et al., 2015), but negative impacts of decreasing pH on phytoplankton growth have also been reported and attributed to pH-induced alterations in algal cell physiology, acid-base chemistry, trace metal availability, ion transport, protein functions, and nutrient uptake (Doney et al., 2009; Gao and Campbell, 2014; Richier et al., 2014; Mackay et al., 2015; Thoisen et al., 2015). Due to these potential antagonist effects, it is still difficult to predict how a specific bloom in a given area will respond to the projected decrease of ocean pH.

A ubiquitous, biogenic trace gas produced in the ocean (Keller et al., 1989; Townsend and Keller, 1996; Kiene et al., 2000), DMS, accounts for 80% of the biogenic sulfur emitted from the ocean to the atmosphere (Kettle and Andreae, 2000). Once released to the atmosphere, DMS undergoes a rapid photochemical transformation into sulfate, which may result in an increase in the concentration of sulfate aerosols and cloud condensation nuclei. Emissions of DMS thus can increase cloud albedo and potentially cool the climate (Ferek et al., 1995; Quinn et al., 2002). The effect of DMS on climate is particularly important in regions of low aerosol burden, such as the summer Arctic atmosphere (Mungall et al., 2016). This gas is mostly produced by the degradation of dimethylsulfoniopropionate (DMSP), a cellular compound present in the majority of phytoplankton species where it fulfils several physiological functions, including osmoregulation (Dickson et al., 1980; Kirst, 1996; Van Bergeijk et al., 2003), cryoprotection (Kirst et al., 1991), and protection against reactive oxygen species (Sunda et al., 2002). A significant fraction of algal DMSP is released to the water column by exudation or from grazing; constituting a dissolved pool that is rapidly consumed by heterotrophic bacteria, which can cleave it into DMS (the "cleavage pathway") or metabolize it via the demethylation/ demethiolation pathway (Kiene et al., 2000). Both unicellular algae and bacteria can thus convert DMSP to DMS, their relative contribution being governed by the taxonomic composition of the phytoplankton community, as well as the abundance and physiological state of the bacteria (Stefels et al., 2007). Oceanic DMS production is thus closely linked to phytoplankton bloom development, which makes it also potentially sensitive to OA.

Several studies have already highlighted the sensitivity of DMS production to decreases in seawater pH. The majority of these experimental studies revealed a negative impact of decreasing pH on DMS production (Hopkins et al., 2010; Avgoustidi et al., 2012; Archer et al., 2013; Webb

et al., 2016), but some have reported either no effect or a positive effect (Vogt et al., 2008; Hopkins and Archer, 2014). Several hypotheses have been proposed to explain these contrasting results. Some authors attribute the pH-induced variation in DMS production to an alteration of the physiological properties of the phytoplankton cells or of the bacterial DMSP metabolism (Vogt et al., 2008; Hopkins et al., 2010, Avgoustidi et al., 2012; Archer et al., 2013; Hopkins and Archer, 2014; Webb et al., 2015, 2016), whereas others evoke an interaction between the DMSP producers and their grazers (Kim et al., 2010; Park et al., 2014). So far, only one study has looked at the impact of OA on DMS dynamics in the Arctic (Archer et al., 2013). The results of this mesocosm study, conducted near Svalbard, show a decrease in DMS concentrations at low pH, suggesting that OA may significantly reduce DMS emissions to the atmosphere in the Arctic.

The main objective of this study was to experimentally assess the impact of decreasing pH on the DMS produced by an Arctic phytoplankton seasonal bloom. Furthermore, we investigated how light conditions corresponding to those experienced by a marginal ice phytoplankton bloom and an under-ice or subsurface phytoplankton bloom could modulate the effects of pH on phytoplankton and DMS variations. The latter was motivated by the strong contribution of subsurface phytoplankton layers to annual productivity in several sectors of the Arctic as well as the apparent increasing occurrence of under-ice blooms, attributed to the thinning of sea ice and the replacement of the multiyear ice by first-year ice (Martin et al., 2010; Frey et al., 2011; Palmer et al., 2014 and references therein).

## 2 Methods

**2.1 Initial water collection and experimental setting**

The incubation experiment was conducted on board of the Canadian research icebreaker *CCGS Amundsen* between 6 and 15 August 2015. The water was collected near the nitracline at 38 m depth at station BB-3 in Baffin Bay (see Fig. 1) using 12-L Niskin-type bottles deployed on a CTD rosette system. Since the cruise took place after the summer bloom in this part of the Arctic, we collected the water between the subsurface chlorophyll maximum (SCM) and the nitracline in order to have sufficient nutrient to support a bloom during our incubation. *In situ* temperature and salinity were respectively -1.35°C and 32.67 at the sampling depth. All water manipulations were done under dim light conditions in order to protect the cells from potential light shock. After the initial collection, water was gravity filtered through a 200 $\mu$m Nitex mesh, in order to remove large grazers, and transferred to 12 gas-tight 10-L bags (HyClone Labtainer©, Thermo Scientific) using a Teflon tube installed between the Niskin bottle and the luer-valve of each bag. All 12 bags were placed in an incubator on the ship's foredeck, through which surface water was circulated to maintain the incubation bags close to sea surface temperature. Since our deck incubator was cooled with circulating surface water, we had no control over the temperature during the incubation (mean temperature of 4.3 ± 1.6°C over the 9-day experiment). However, all bags were in the same incubator, hence submitted to the same temperature. An insulated box (a "cooler") was used to transport the samples whenever they had to be carried between the incubator and the ship-board laboratory. Water temperature was monitored every 15 minutes during the entire experiment using a RBR TR-1060 temperature recorder. Incident photosynthetically active radiation (PAR, 400-700 nm) was monitored continuously using a LI-1000 Data logger combined with a LI-190SA cosine sensor (LICOR) located near the incubator. Incident UVA (SED033 detector/UVA filter/W diffuser) and UVB (SED240 detector/UVB filter/W diffuser) radiation was measured daily (day 5 to 9 only due to instrument dysfunction during the first four days) around midday with an IL1700 radiometer (International Light).

## 2.2 Treatments and acidification protocol

*Acidification protocol*

For this experiment, the phytoplankton communities were exposed to a pH gradient and two light regimes. Twelve incubation bags were separated into 2 groups of six bags to produce two similar sets of pH gradients (control + 5 acidified bags; Table 1). The pH gradient method has been successfully applied by Schulz et al. (2013) and Paul et al. (2016) and is suitable when the possibility of replication is limited (see Cottingham et al., 2005, and Havenhand et al., 2010, for more details).

After equilibrating at incubator temperature, the "non-control" samples were taken to the lab and acidified (at time T0, 6 August) by addition of strong acid and bicarbonate (HCl, 0.02 N; $NaHCO_3$, 0.3 N) following procedures described by Riebesell et al. (2010). By keeping the alkalinity constant, this procedure mimics the natural acidification process as observed in global ocean waters. The volumes needed to reach each targeted pH value on the total hydrogen ion scale (mol $kg^{-1}$ SW) were determined with the help of the MS Excel macro $CO_2SYS$ (Pierrot et al. 2006) (using the carbonic acid dissociation constants ($K_1$ and $K_2$) of Luecker et al., 2000 and the bisulfate dissociation constant ($K_{HSO4}$) of Dickson, 1990). The acid and bicarbonate solutions were added in appropriate proportions to each bag using a syringe connected to the Luer-lock port of the bag. After acidification, each bag was gently inverted around 10 times before being returned to the incubator. The entire process of the bags acidification was completed within 4 hours. The incubator was then covered with a white canvas sheet until sunset and bags remained overnight in the incubator without further manipulation. After the initial acidification, the carbonate chemistry was not manipulated and the only changes in pH that occurred during the experiment were due to biological activity. Subsampling began the next day in the morning (T1, 7 August, around 7:00 am).

*Light treatments*

Once acidified, each set of incubation bags covering the initial pH values between 8.1 and 7.2 was assigned to a light treatment (Tables 1 and 2): "low light" (LL, 6 bags), and "high light" (HL, 6 bags). The transmittance through the incubation bags of both LL and HL treatments was determined in the laboratory using a Perkin Elmer LAMBDA 850 Spectrophotometer. We based our experimental "low light" conditions (PAR, UVA and UVB transmittance of 32.6%, 20.6%

and 0%, respectively) on a model of the minimal light conditions required to trigger an under-ice bloom (Palmer et al., 2014) and available data from measurements of UV and PAR under ponded ice (Trodahl and Buckel, 1990; Perovich et al., 1998; Belzile et al., 2000; Frey et al., 2011; Palmer et al., 2014). As UVB is strongly attenuated by sea ice and thus likely negligible at 5 m depth under the ice (Frey et al., 2011; Palmer et al., 2014), those short wavelengths were

eliminated. The LL treatment transmittances were achieved by covering the 6 LL bags with 2 layers of Nitex mesh (300 $\mu$m) and one layer of Mylar D film (0.13 mm thick; Demers et al., 1998). The Nitex mesh attenuated the entire light spectrum equally whereas the Mylar D film specifically removed the leftover UVB radiation (Demers et al., 1998; Roux et al., 2002). The resulting transmittance measured through the incubation bags under LL treatment was similar to

light conditions encountered at 5 m depth under the ice in early summer (Frey et al., 2011) or at subsurface chlorophyll maxima, which are widespread in the Arctic Ocean (Martin et al. 2010, 2013).

The incubation bags dedicated to the HL treatment (PAR, UVA and UVB transmittance of 77.5%, 61.5% and 32.6%, respectively) were exposed to light conditions representing a typical

Arctic upper mixed layer of approximately 5 m depth, based on the equation of Riley (1957):

$$E_z = E_0 \cdot \frac{1 - e^{-K_d \cdot z}}{K_d \cdot Z}$$

where $E_z$ ($\mu$mol quanta m$^{-2}$ s$^{-1}$) is the daily PAR averaged over the surface mixed layer, $E_0$ ($\mu$mol quanta m$^{-2}$ s$^{-1}$) is the daily averaged PAR at the surface, $K_d$ is the PAR diffuse attenuation coefficient (assumed to be 0.123 m$^{-1}$; unpublished data from Baffin Bay, M. Gosselin), and Z (m)

is the surface mixed layer depth. The attenuation of incident light for the HL treatment was only due to the bag itself.

**2.3 Microcosm subsampling for chemical and biological variables**

The 12 incubation bags (microcosms) were monitored over 9 days after the initial acidification (T0). Subsampling took place at 9:00 am the first day (T0), before the initial acidification, and

205 between 6:00 and 10:00 a.m. for the 9 following days (T1-T9). Not all the variables were sampled every day. Water for pH, total alkalinity (TA), dissolved inorganic carbon (DIC), and DMS analyses was collected directly from the microcosms to minimise outgassing, whereas

water for the measurement of chlorophyll *a* (Chl *a*), nutrients, flow cytometry, and taxonomy was first collected in a brown bottle (Nalgene®, 1L) and kept at 4°C in the refrigerator until subsampling, generally 2 h after initial water sampling. Water for salinity determination was taken at T0 before the acidification process and at T9 in each incubation bag. Samples for salinity were stored in the dark in 250 mL plastic bottles until analysis in May 2016 using a Guidelines 8400B salinometer. At least half of the initial volume of the microcosms (5L) remained in the bags at the end of the experiment.

*Carbonate system*

The pH was measured on T1, T2, T4, T6, T8, and T9 using an Agilent 8453 UV/Vis Spectrophotometer on the total hydrogen ion scale ($pH_T$). Water was directly subsampled from the incubation bags to the spectrophotometric cells in order to minimise gas exchange with ambient air. The spectrophotometer cells (10 cm path length, V = 33 mL) were brought up to 25°C in an aluminum block before readings on the spectrophotometer (approximately 1 hour after sampling). Absorbance was measured at 434, 578, and 730 nm before and after addition of 50 $\mu$L of a *m*-cresol purple indicator dye solution (Clayton and Byrne, 1993). A second addition of 50 $\mu$l of *m*-cresol purple indicator dye solution was made to the sample in order to determine the extent of the dye perturbation on $pH_T$ values. Duplicates were analysed for each incubation bag, and $pH_T$ subsequently calculated according to Dickson et al. (2007, see their section SOP 6b). Data were then converted to mean incubation temperature using $CO_2SYS$ and the TA measurements (Pierrot et al., 2006). The $pCO_2$ values corresponding to each $pH_T$ measurement were also calculated using $CO_2SYS$ software. Concentrations in protons for each microcosm were calculated using the $pH_T$ measurements.

Water for TA and DIC analyses was subsampled early in the morning on days T1, T4 and T9. Water was gently collected from the incubation bags in pre-rinsed 250 mL glass bottles. Samples were analysed on board within 4 hours by coulometric and potentiometric titration for DIC and TA, respectively (more details are given for example in Johnson et al., 1993, or Dickson et al., 2007), using a VINDTA 3C (Versatile Instrument for the Determination of Titration Alkalinity, Marianda). Both the DIC and TA measurements were calibrated against Certified Reference Materials (CRMs, Andrew Dickson, Scripps Institution of Oceanography, USA) and the

reproducibility was better than 1 and 2 $\mu$mol kg$^{-1}$ SW, respectively, for the DIC and TA measurements.

*Nutrients*

Water for nutrient analysis was filtered through Luer-lock syringe acrodisc filters (GMF, porosity of 0.7 $\mu$m) into 15 mL acid-washed polyethylene tubes. After collection, samples were stored at 4°C in the dark and analysed within 1 hour for nitrate ($NO_3^-$) plus nitrite, nitrite ($NO_2^-$), silicic acid ($Si(OH)_4$), and soluble reactive phosphate (SRP) on a Bran and Luebbe Autoanalyser III using a colorimetric method adapted from Hansen and Koroleff (2007) (detection limits for; $NO_3^-$

: 0.03 $\mu$mol L$^{-1}$; $NO_2^-$: 0.02 $\mu$mol L$^{-1}$; $Si(OH)_4$: 0.1 $\mu$mol L$^{-1}$; SRP: 0.05 $\mu$mol L$^{-1}$).

*Plankton biomass and enumeration*

Water for Chl *a* concentration analysis was subsampled from each incubation bag every day using 1 L brown polyethylene bottles. The bottles were stored at 4°C in the dark for 2 hours before filtration onto 25 mm Whatman GF/F filters (0.7 $\mu$m nominal pore size). Phytoplankton

pigments were extracted in 90% acetone and stored at 4°C in the dark for 20 hours. The fluorescence of the extracted pigments was then measured using a Turner Designs fluorometer 10-AU after acidification according to the method described by Parsons et al. (1984). Chl *a* concentrations were calculated from the equation published in Holm-Hansen et al. (1965).

For the enumeration of phytoplankton ($\leq 20$ $\mu$m) and bacteria, sterile cryogenic polypropylene

vials were filled with 4 mL of water to which 20 $\mu$L of glutaraldehyde (Grade I, 25% in water, Sigma Aldrich; Marie et al., 2005) were added. Duplicate samples were flash frozen in liquid nitrogen after standing 15 min at room temperature in dark. These samples were then stored at -80°C for 5 months until flow cytometry analysis. After defrosting to ambient air temperature, samples were analysed using a FASC Calibur FCB3 flow cytometer (Becton Dickinson). One of

the duplicate samples was used to determine the abundance of picophytoplankton (0.2-2 $\mu$m), which include picoeukaryotes and picocyanobacteria, and nanophytoplankton (2-20 $\mu$m) based on theirs autofluorescence characteristics and size (Marie et al., 2005). The other tube was used for bacterial counts (Marie et al., 1999).

On days T0, T5 and T9, water samples (between 100 and 250 mL depending on the biomass)

were taken for identification and counting of eukaryotic cells (protists) larger than 2 $\mu$m. Samples

were preserved in acidic Lugol's solution (Parsons et al., 1984) and stored in the dark for 7 months until enumeration with an inverted microscope (WILD Heerbrugg) equipped with phase contrast optics (Lund et al., 1958). Thirteen selected samples collected under both light treatments and representing the whole range of pH amendments (i.e. $pH_T$ 8.1 (control), 7.6 and 7.2) were counted.

*Fast Repetition Rate Fluorometry (FRRF) measurements*

For FRRF measurements, 20 mL of seawater were taken directly from the microcosms at $pH_T$ 8.1, 7.8, and 7.2 for both HL and LL treatments on days T2, T4, T6, T7 and T9. Water was subsampled early in the morning in order to avoid photoinhibition (Schuback et al., 2015). The FRRF measurements were conducted on a bench top FRRF instrument (Soliense Inc.), as described in Schuback et al. (2015). For each sample, background fluorescence blanks were prepared by syringe filtering a small amount of water through a Whatman GF/F filter. Samples were kept at very low light (approx. 5 $\mu$mol quanta m$^{-2}$ s$^{-1}$) at *in situ* temperature for 45 minutes before the measurements. A single turnover (ST) excitation protocol was applied to derive minimum ($F_o$) and maximum ($F_m$) chlorophyll *a* fluorescence yields by applying an iterative non-linear fitting procedure to a mean of 20 consecutive ST flashlet sequences using custom software (Kolber et al., 1998). The $F_o$ and $F_m$ were used to calculate $F_v/F_m$ as $(F_m-F_o)/F_m$. In the absence of iron limitation, this ratio can be used as a sensitive indicator of algal photosynthetic efficiency (Maxwell and Johnson, 2000; Schuback et al., 2016).

*DMSP and DMS concentrations*

For the quantification of dissolved DMSP ($DMSP_d$), 20 mL of the water sample were gravity-filtered through 47 mm Whatman GF/F filters and the first 3.5 mL of filtrate were stored in 5 mL polyethylene tubes using the less disruptive Small-Volume gravity Drip Filtration (SVDF) method described by Kiene and Slezak (2006). Tubes for total DMSP ($DMSP_T$) samples were directly filled with 3.5 mL of unfiltered water. Samples of both $DMSP_d$ and $DMSP_T$ were preserved by adding 50 $\mu$L of a 50% sulfuric acid ($H_2SO_4$) solution to the 5 mL tubes, which were then stored at 4°C in the dark until analysis in laboratory. DMSP concentrations were determined as DMS through hydrolysis with a 5 N solution of NaOH, purging and cryotrapping in liquid nitrogen, followed by analysis on a Varian 3800 Gas Chromatograph equipped with a Pulsed Flame Photometric Detector (Varian 3800) with a detection limit of 0.1 nmol L$^{-1}$ (Scarratt

et al., 2000; Lizotte et al., 2012; Galindo et al., 2016). All DMSP samples were calibrated against multiple micro-injections of a 100 nmol $L^{-1}$ solution of standardized dimethyl-B-propiothetin (DMPT from Research Plus) hydrolyzed with a 5 N solution of NaOH.

Water samples for DMS concentration analysis were withdrawn on days T0, T2, T4 and every day for the remainder of the incubation period. DMS concentrations were determined onboard the ship using purging, trapping and S-specific gas chromatography, as described by Asher et al. (2015). The custom-built system described by Asher et al. (2015) was used here in manual mode by directly injecting 10 mL of each incubation sample into a sparge vessel. Water for DMS analysis was subsampled directly from each bag using a Luer-lock syringe and injected into the custom-built system at most 40 minutes after sampling. Ultra-high purity $N_2$ was used to extract the volatile DMS from solution at a flow rate of 100 mL $min^{-1}$. The sparged DMS was adsorbed onto Carbopack-X packed in a stainless steel trap. After trapping was completed, a series of high current pulses heated the trap to 250°C to desorb the DMS onto the capillary column prior to elution and detection by a Pulsed Flame Photometric Detector (PFPD, OI Analytical, Model 5380). Light emitted during combustion in the PFPD was converted to a voltage (using a PMT – photomultiplier tube) and recorded by the LabView software. Raw data outputs (peak voltages) were processed using Matlab by calculating the area under each curve and by comparing them to standard curves to give final DMS concentrations. Known DMS concentrations of 0, 3, 6, 9, 12, and 15 nmol $L^{-1}$ were processed for calibration purpose before each sampling day in order to obtain the standard curves. Area under each standard curve was calculated using Matlab and associated with its known DMS concentration.

### 2.4 Statistical analysis

All statistical analyses were run using RStudio (http://www.rstudio.com/). Normality of the data was determined using a Shapiro-Wilk test at the 0.05 significance level and data were transformed (log or square root) when the normality was rejected ($p < 0.05$). To test for differences between pH and light treatments for the variables measured during the experiment (time-series of nutrient and Chl $a$ concentrations, phytoplankton, bacteria abundances, and dimethylated compound concentrations), we used a generalized least square model that corrected for data autocorrelation due to time repetition (gls command in R studio, package nlme; also see the method described by Paul et al., 2016). In the model, time and pH were taken as two

continuous factors; the light was included as a categorical factor (2 levels: "Low Light", "High Light"). As the inclusion of the interaction between light and pH in our gls model did not yield significant results, it was not included in our statistical analysis. Relationships between the average values of all response variables and the set of $pH_T$ were evaluated using the Pearson's linear regression ($r^2$).

# 3 Results

## 3.1 Irradiance conditions

Daily mean incident PAR varied between 189 and 492 $\mu$mol m$^{-2}$ quanta s$^{-1}$ during the 9 days of the incubation. Accordingly, daily mean PAR in the HL and LL treatment varied between 146 and 312 $\mu$mol m$^{-2}$ quanta s$^{-1}$ and between 61 and 131 $\mu$mol m$^{-2}$ quanta s$^{-1}$, respectively (Table 2). Incident UVA and UVB levels varied respectively between 4 and 13 W m$^{-2}$, and between 0.1 to 0.3 W m$^{-2}$ during the last 5 days of the experiment.

## 3.2 State of the carbonate system

Table 1 summarizes values of four carbonate system parameters and associated protons concentrations (pH$_T$, [H$^+$], DIC, TA, $p$CO$_2$) for each light and pH treatment at T0, T1, T4, and T9. At T1, the mean TA in the acidified bags was 2221 $\pm$ 16 $\mu$mol kg$^{-1}$ SW, approximately 1% lower than the value of 2243 $\mu$mol kg$^{-1}$ SW measured in the original sample recovered from 38 m depth at station BB-3. At the beginning of the experiment (T1), the mean pH$_T$ at 4.27°C (mean incubator temperature during the experiment) between HL and LL controls was 7.939 $\pm$ 0.003 (see Table 1 and Fig. 2). Values for the following decreasing pH levels were 7.779 $\pm$ 0.012, 7.641 $\pm$ 0.007, 7.460 $\pm$ 0.001, 7.323 $\pm$ 0.030, and 7,159 $\pm$ 0.004. As expected, the pH$_T$ in all bags increased during the incubation period due to photosynthesis (Rost et al., 2008; Paul et al., 2016) (mean protons change per bag of -1.52 $\times$ 10$^{-8}$ $\pm$ 6.67 $\times$ 10$^{-9}$ mol L$^{-1}$ over the 9-day experiment; Fig. 2). A difference in pH$_T$ (corresponding to an average proton difference of -8.87 $\times$ 10$^{-9}$ $\pm$ 8.59 $\times$ 10$^{-9}$ mol L$^{-1}$) was still observed at T9 between bags in both the LL and HL treatments (Fig. 2). The pH$_T$ gradients between the two light treatments were thus very similar, allowing us to clearly discriminate between pH$_T$ and light effects. Hereafter, we refer to the pH treatments by their mean pH$_T$ values measured over the 9 incubation days: 8.1 (control) and decreasing mean pH$_T$ values of 7.9, 7.8, 7.6, 7.4 and 7.2.

## 3.3 Dissolved inorganic nutrient concentrations

At T1, NO$_3^-$, Si(OH)$_4$, and SRP concentrations varied between 5.07-5.83, 8.87-10.45, and 0.94-1.00 $\mu$mol L$^{-1}$, respectively (Fig. 3). All three nutrients displayed the same general temporal pattern during the incubation, irrespective of the light treatment. Their concentrations remained constant or increased slightly during the first three days and then decreased rapidly with the onset

of the bloom to very low or undetectable concentrations between T6 and T8 for $NO_3^-$ (all $pH_T$) and at T8 for $Si(OH)_4$ (highest $pH_T$ only). Likewise, SRP concentrations decreased in parallel with $NO_3^-$ and $Si(OH)_4$, but remained higher than 0.3 $\mu$mol $L^{-1}$ during the whole experiment. The impact of pH on nutrient consumption was apparent under both light treatments, with nutrients

being generally consumed more rapidly at high $pH_T$ than at low $pH_T$ (Fig. 3, Table 3). The $NO_3^-$ depletion was temporally delayed in the most acidified treatments, reaching below detection values one day after the other treatments for microcosms at $pH_T$ 7.4 (T7) and two days after the others for microcosms at $pH_T$ 7.2 (T8). With the exception of microcosms at $pH_T$ 8.1 under LL, and at $pH_T$ 7.8 under both light treatments, $Si(OH)_4$ remained at relatively high concentrations in

all treatments after the peak of the bloom (Figs. 3c, d and 4a, b). While the ratio of $NO_3^-$ to SRP uptake ($\Delta NO_3^-:\Delta SRP$) during the incubation showed no statistical difference between the six $pH_T$ treatments tested, the ratio $\Delta Si(OH)_4:\Delta NO_3^-$ decreased linearly with increasing proton concentration for both light regimes ($r^2 = 0.72$, $p < 0.001$; data not shown). This change was driven mostly by a reduction in $Si(OH)_4$ uptake, which exhibited a ca. 3-fold difference between

experimental pH extremes compared to a ca. 1.7-fold difference for $\Delta NO_3^-$ uptake (not shown).

**3.4 Phytoplankton biomass**

The mean Chl $a$ concentration before the beginning of the experiment (T0) was $0.686 \pm 0.004$ $\mu$g $L^{-1}$, and increased exponentially from T1 to T5-T8, depending on the $pH_T$ treatment, to reach maximum values varying between 4.5 and 7.5 $\mu$g $L^{-1}$ (Fig. 4a, b). The bloom period was followed

by either a plateau or a small decrease in Chl $a$ biomass. The temporal changes in Chl $a$ concentrations were not affected by the light regimes but we observed a significant difference in Chl $a$ concentrations between $pH_T$ treatments (Table 3). Figure 5a shows the relationship between the mean Chl $a$ concentrations and the corresponding mean proton concentrations over the entire experimental period. For both light regimes (data pooled), the mean Chl $a$ concentration

decreased linearly with increasing proton concentration (Fig. 5a, Table 4).

**3.5 Phytoplankton abundance and taxonomy of the protist community > 2 $\mu$m**

Nanophytoplankton (algal cells between 2 and 20 $\mu$m) abundance varied during the incubation from $0.81 \times 10^3$ cells $mL^{-1}$ on day T1 to $19 \times 10^3$ cells $mL^{-1}$, a maximal value reached at $pH_T$ 7.61 on day T8 under LL conditions (Fig. 4c,d). For both light treatments, nanophytoplankton

abundance increased steadily from T1 to T6-T8 and decreased or remained stable thereafter. This

trend is comparable with the trend observed for Chl $a$ (Fig. 4a,b) but with the slight difference that the nanophytoplankton generally reached their maximal abundance 1-2 days after the Chl $a$ concentration maxima. The lag probably reflects a decrease in Chl $a$ synthesis (and thus Chl $a$ cell quota) as the dividing cells were becoming nitrogen limited. Variations in nanophytoplankton abundance were significantly correlated with Chl $a$ during the growth phase ($r^2 = 0.78$, p < 0.001), suggesting that nanophytoplankton were responsible for most of the biomass build-up during the bloom. Like Chl $a$, only the $pH_T$ gradient (not light treatment) appeared to have affected nanophytoplankton dynamics as we observed significant differences between nanophytoplankton abundances between all $pH_T$ tested (Table 3). As shown in Figure 5b, the mean nanophytoplankton abundances decreased as the proton concentration increased (data pooled, Table 4).

The initial protist community > 2 $\mu$m was dominated by centric diatoms (33% of total protist abundance), followed by cryptophyceae (22%), unidentified flagellates (17%), and prasinophyceae (12%) (Fig. 6). Pennate diatoms, dinophyceae and chrysophyceae were also present in smaller proportions (4, 3 and 2% respectively; included in the group named "others" in Fig. 6, given their low abundance). Among the protists > 2 $\mu$m, centric diatoms were dominant in most treatments at T5 and T9, accounting for between 72% and 89% of total cells, while the other groups initially present decreased in abundance or became undetectable. More than 90% of the centric diatoms belonged to the genus *Chaetoceros*. Although the dominant species of *Chaetoceros* could not be identified (the biomass of this genus was dominated by unidentified *Chaetoceros* comprised between 2 and 5 $\mu$m), *Chaetoceros gelidus*, *Chaetoceros wighamii*, *Chaetoceros cf. karianus* and *teniussimus* also contributed to the bloom. All microcosms subsampled during the experiment showed very similar taxonomic compositions, with *Chaetoceros* spp. dominating the community and accounting for more than 65% of cell abundance. The sole exception was the LL control microcosm ($pH_T$ of 8.1), in which the species composition on day T9 was more evenly distributed, with 58% of cell abundance attributed to groups other than centric diatoms. Within this fraction, flagellates accounted for 24% of the total abundance, choanoflagellidea for 12%, prasinophyceae for 10%, cryptophyceae for 2%, and minority protist groups ("others") for 11%. Due to the low number of samples analyzed by inverted microscopy, it was not possible to statistically confirm whether the difference observed on day T9 between the control microcosm and the two other microcosms analyzed was related to

the pH gradient or resulted from inter-microcosm variability. Nevertheless, the general trends visible in Figure 6 suggest that the relative abundance of each protist group was not influenced by the light regime or the pH gradient. Centric diatoms, and especially *Chaetoceros* spp., remained the dominant taxa throughout the incubation period in every microcosm. Dinophyceae, which initially contributed up to 3% of the protist abundance, became undetectable in all microcosms, including the controls, suggesting a negative effect of bag manipulation and the incubation process on this taxon during the experiment.

During the present experiment, picophytoplankton (algal cells between 0.2 and 2 $\mu$m) accounted for more than 90% of total phytoplankton cells ≤ 20 $\mu$m. Photosynthetic picocyanobacteria made up ≤ 2% of total abundance of picophytoplankton (i.e. picocyanobacteria + photosynthetic picoeukaryotes). Picophytoplankton cell abundance was low at the start of the experiment ($< 5 \times 10^3$ cells mL$^{-1}$) and remained at this low level until T3 when it began to increase to reach maximum concentrations (200-300 $\times 10^3$ cells mL$^{-1}$) between T7 and T8, depending on the pH$_T$ (Fig. 4e, f). Exceptions to this trend are the two microcosms at the lowest pH$_T$ under both light regimes, in which picophytoplankton abundance continued to increase linearly until the end of the experiment. Of the two stressors tested (light and pH), only pH$_T$ had a weak but significant effect on picophytoplankton abundances (Table 3). As shown in Figure 5c, mean picophytoplankton abundance increased with increasing proton concentration for both light treatments although this relation does not appear to be linear (data pooled, Table 4).

**3.6 Photosynthetic performance**

The F$_v$/F$_m$ ratio, a parameter widely used to indicate the photosynthetic performance of primary producers, was very similar between the two light treatments at each sampling day (Table 4). On days T2 and T4, the F$_v$/F$_m$ ratios were higher than 0.5 under both light treatments and pH$_T$ values of 8.1, 7.8, and 7.2, suggesting that the blooming algal cells were in good physiological condition. The ratios then decreased to ca. 0.3-0.4 between days T6-T9, when NO$_3^-$ started to be exhausted in most microcosms and the increase in Chl *a* biomass and phytoplankton cell abundance had stopped (Figs. 3a, b and 4a-f). These results show low photosynthetic performance in the microcosms at the end of the experiment.

### 3.7 Bacteria

Bacterial abundances under both light treatments showed a moderate increase during the experiment (Fig. 4g,h). Although it is not evident on Figure 4 (g, h), the bacterial abundances were significantly different between both light and pH treatments, in contrast with the other variables presented above (Table 3). As shown in Figure 5d, the mean bacterial abundance decreased with increasing proton concentrations under HL but showed no proton/pH related effect under LL (Table 4).

### 3.8 DMSP and DMS concentrations

The $DMSP_T$ concentrations showed the same general temporal pattern in all microcosms (Fig. 7a,b). Initial average $DMSP_T$ concentration was $7 \pm 3$ nmol $L^{-1}$, and remained at this low level during the first five days of the experiment, increased sharply between T5 and T6 to reach values of 80 nmol $L^{-1}$ (under HL) and 120 nmol $L^{-1}$ (under LL), and remained high but variable for the rest of the experiment. Of the two stressors tested, only the pH gradient significantly influenced the $DMSP_T$ concentrations (Table 3). As shown in Figure 8a, the mean $DMSP_T$ concentration decreased with increasing proton concentration, but only under the HL treatment between $pH_T$ 8.1 – 7.6 (Table 4).

The $DMSP_d$ concentrations started at 0.5-1.8 nmol $L^{-1}$, decreased slightly until T5 and then tended to increase, again slightly, to reach maximum values between 2 and 3 nmol $L^{-1}$ near the end of the experiment (Fig. 7c,d). Neither the light nor the $pH_T$ treatments significantly affected the $DMSP_d$ concentrations over the period of incubation (Table 3).

In most of the microcosms, temporal changes in DMS concentrations were very similar to those in $DMSP_T$ and were characterised by a gradual increase from 0.54 to ca. 4 nmol $L^{-1}$ between T1 and T5, a sudden increase to 10-20 nmol $L^{-1}$ between T5 and T7, followed by either a plateau or a decrease in concentrations until the end of the experiment (Fig. 7e,f). The sharp increase in DMS was not observed at the two lowest levels of $pH_T$ (i.e. 7.4 and 7.2). At these low $pH_T$ values, DMS concentrations did not exceed 6 nmol $L^{-1}$ during the phytoplankton growth phase. Of the two stressors tested, only the pH gradient had a significant effect on the DMS concentrations over time (Table 3). As shown in Figure 8b, the mean DMS concentration decreased with increasing proton concentration (Table 4). The average $DMS:DMSP_d$ ratio followed the same trend as for

DMS concentrations, showing a linear decrease over the whole range of proton concentration,

irrespective of the light treatment (Fig. 8c, Table 4).

## 4 Discussion

In this study, a natural Arctic plankton community in a pre-bloom stage (initial high nutrient-low Chl *a* concentrations) was exposed over 9 days to reduced pH conditions under two contrasting light regimes. The two light regimes were designed to simulate the mean irradiance in an ice-free 5-m thick surface mixed layer (HL, marginal ice bloom conditions) and the mean irradiance at 5 m depth under a melting ponded ice pack (LL, under-ice bloom/ subsurface chlorophyll maximum conditions). The $pH_T$ gradient comprised 6 levels covering the range of pH expected between the present and the year 2300. We recognize that the rapid change in pH to which the plankton assemblage was exposed at the beginning of our study is not representative of the more gradual acidification that is taking place in the ocean. For this reason, negative impacts should be interpreted as potential extreme responses, and the planktonic community could be more adaptable than is implied by our results.

### 4.1 General bloom characteristics and associated variations in DMSP and DMS

During this experiment, a phytoplankton bloom numerically dominated by the centric diatom *Chaetoceros* spp. and by photosynthetic picoeukaryotes occurred almost simultaneously in all microcosms (Figs. 4c-f). The exponential growth phase lasted 6-7 days, a period during which the concentration of all three monitored nutrients decreased, with $NO_3^-$ reaching near-zero values. The $F_v/F_m$ ratios around 0.5 characterised the period of exponential growth in all microcosms, including the controls (Table 5). The peak of the bloom and $NO_3^-$ depletion were followed by a decrease of $F_v/F_m$ ratios and a stabilization or slight decrease in phytoplankton abundances (Figs. 3 and 4c-f). As the initial concentrations of nutrients in the incubation bags were similar to the concentrations found in the upper mixed layer before the spring bloom in Baffin Bay, we are confident that the bloom that took place in our bags is comparable to the spring bloom naturally developing in these waters (Tremblay et al., 2002, 2006). Furthermore, the dominance of diatoms during our experiment was expected as blooms of this phytoplankton group are commonly found at the marginal ice zone in the Arctic, and in Baffin Bay in particular (Poulin et al., 2011), and tend to dominate spring primary production when nutrients are non-limiting in this region (Heimdal, 1989; Matrai and Vernet, 1997; Wolfe et al., 1999; Von Quillfeldt, 2000).

Picophytoplankton were numerically more abundant than nanophytoplankton during our experiment, representing between 54% (during the diatom bloom) and up to 96% (post bloom

period) of total phytoplankton abundance. High picophytoplankton abundances (64% of total photosynthetic cell abundance $\leq 20$ $\mu$m) have been reported by Tremblay et al. (2009) in the Canadian Arctic where, due to their small size, they accounted for only 16% of total Chl $a$. During the growth phase of this study, concentrations of Chl $a$ displayed a stronger correlation with nanophytopankton abundances ($r^2 = 0.78$, p < 0.001) than they did with picophytoplankton ($r^2 = 0.48$, p < 0.001) despite the latter's numerical dominance. The taxonomy of the picophytoplankton was not determined during our study, but *Micromonas*-like species are known to be a major component of the photosynthetic picoeukaryotes in the Arctic (Lovejoy et al., 2007; Tremblay et al., 2009).

The $DMSP_T$ concentrations measured during the development of the diatom bloom correspond to a $DMSP_T$:Chl $a$ ratio ranging from 3 to 15 nmol $\mu$g$^{-1}$ (data not shown), as expected for this phytoplankton group (Stefels et al., 2007). The sharp increase in $DMSP_T$ measured between T5 and T6 under both light regimes and at all $pH_T$ values investigated was however unexpected (Fig. 7a,b), as it contrasts with results from previous mesocosms experiments that showed a more gradual increase in DMSP along with algal biomass (Vogt et al., 2008; Archer et al., 2013; Park et al., 2014). During our experiment, this sharp increase in $DMSP_T$ and DMS coincided with the exhaustion of $NO_3^-$, with the exception of the microcosms at $pH_T$ 7.4 and 7.2 under both light regimes (Fig. 3a, b). At the two latter pH the increase of $DMSP_T$ between T5 and T6 was of lower magnitude compared to the other microcosms and the remaining $NO_3^-$ concentrations varied from 0.9 to 2.6 $\mu$mol L$^{-1}$ in these bags at T6 (Fig. 7a, b). Previous laboratory experiments have shown that $NO_3^-$ limitation could induce a 25-fold increase in diatoms $DMSP_T$ cellular quotas (Keller et al., 1999; Bucciarelli and Sunda, 2003; Sunda et al., 2007). Our results show a ca. 9-fold increase of $DMSP_T$ between T5 and T6, which corresponds well with the values found in the literature (Sunda et al., 2007). Unicellular algae are known to synthesize DMSP during episodes of stress or senescence as an overflow mechanism to evacuate excess energy, sulfur, and carbon, while allowing the cell to function (Stefels et al., 2000; Hopkins and Archer, 2014). In response to $NO_3^-$ limitation, diatoms could have switched from the synthesis of glycine betaine (GBT), a nitrogen-containing osmolyte, to its sulfur analog DMSP (Andrea, 1986; Keller et al., 1999). Altogether, these results suggest a relationship between the intensity of nitrate depletion and the magnitude of DMSP synthesis by the diatom community during our experiment. These results also suggest that diatoms could have more difficulty in efficiently taking up/assimilating $NO_3^-$ at lower pH.

**4.2 Phytoplankton community and nutrient uptake response to the pH gradient**

Lowering the pH had a negative impact on the mean concentration of Chl $a$, as well as on the mean abundance of nanophytoplankton during the 9-day experiment (Fig. 5a,b, Table 4). Considering that *Chaetoceros* diatoms accounted for more than 65% of the total abundance of the large phytoplankton ($> 2$ $\mu$m) and most of the Chl $a$ build-up, these results suggest that net carbon fixation by diatoms was negatively impacted by the decrease in pH, as observed in other studies (Gao and Campbell, 2014, and references therein). Altered seawater carbonate chemistry could perturb the energy requirements of diatom cells, leading to changes in respiration, cell surface and intracellular pH stability (Gao and Campbell, 2014 and reference therein). Such energy re-allocation could force diatoms to assign more energy to repair mechanisms and ion transport to remedy the acid-base perturbation, thus impairing their growth.

The relative uptake of nutrients was also impacted by the $pH_T$ treatments in diatoms. The observed linear decrease of the $Si(OH)_4$: $NO_3^-$ concentration ratio with increasing proton concentration, which was mostly driven by a reduction in $Si(OH)_4$ consumption, either suggests that, at low $pH_T$, diatoms cells were less silicified or that non-diatom phytoplankton made a larger contribution to nutrient drawdown. The former is the most likely explanation since diatom abundance increased in all experimental treatments and numerically dominated the cells community $> 2\mu$m at T5 and T9, irrespective of the $pH_T$ level tested. This explanation is also consistent with laboratory studies that previously reported an impairment of $Si(OH)_4$ uptake in diatom cultures grown at low pH (Milligan et al., 2004; Hervé et al., 2012; Mejía et al., 2013). In Hervé et al. (2012), the cells exhibited similar growth rates at high and low pH, implying that the negative impact of a low pH on silification does not preclude bloom development in nature.

The picophytoplankton were also impacted by the decrease in $pH_T$ (i.e. the augmentation of proton concentration), albeit differently from the nanophytoplankton. As shown in Figure 5c, their mean abundance increased as the proton concentration increased from 0.1 to $2.0 \times 10^{-8}$ mol $L^{-1}$ and then tends to stabilise at $pH_T$ below 7.6 (see Table 1 for the corresponding proton concentration). In contrast to our study, Richier et al. (2014) reported a negative impact of ocean acidification not only on nanophytoplankton but on picophytoplankton as well during a microcosm experiment using a similar methodology. In this study conducted with water from the northwest European shelf, lowering the pH resulted in a decrease in the abundance (cell number)

and biomass (Chl *a*) of phytoplankton < 10 $\mu$m. These contrasting results could reflect differences in the initial picophytoplankton community composition and possible species-specific physiological response to OA. By contrast, a positive influence of a decreasing $pH_T$ on picophytoplankton abundance, and particularly *Micromonas*-like phylotypes, has been previously reported by Hama et al. (2016) for a coastal planktonic community near Japan, as well as by others based on mesocosm experiments (Paulino et al., 2008; Brussaard et al., 2013). This response could be explained by an adjustment of the CCM (Carbon Concentrating mechanism) used by these small cells at low pH. Indeed, picophytoplankton may rely more on $CO_2$ diffusion at lower pH than on investing energy in active $CO_2$ and $HCO_3^-$ uptake (Brussaard et al., 2013). The energy saved that way could translate into higher growth. Moreover, small picophytoplankton species are known to possess less effective CCM than diatoms (Mackey et al., 2015). If the picophytoplankton thriving in our experiment were not saturated with $CO_2$ at *in situ* levels, an increase in $pCO_2$ could have stimulated photosynthesis and growth of this size group (Brussaard et al., 2013).

In summary, during our experiment, OA stimulated picophytoplankton net growth over the whole range of $pH_T$ investigated while impairing the development of nanophytoplankton, especially at the lowest $pH_T$ tested. This implies that OA will most probably maintain or increase the numerical dominance of picophytoplankton over the nanophytoplankton in the Arctic Ocean (Newbold et al., 2012; Davidson et al., 2016). Tremblay et al. (2012) suggested that climate warming and the associated increase in surface water stratification could also favour the growth of small cells relative to larger diatoms, thus favouring the diminution of primary producers biomass. Our results suggest that OA could accentuate this community response caused by warming and stratification.

**4.3 Effect of the pH gradient on dimethylated compounds**

During our experiment, $DMSP_T$ and DMS responded differently to decreasing $pH_T$. While $DMSP_T$ concentrations decreased between $pH_T$ of 8.1 and 7.6 (see Table 1 for corresponding proton concentration values) only under HL conditions, DMS concentrations decreased linearly over the full range of $pH_T$ investigated and under both light regimes (Fig. 8, Table 4).

Although not linear over the full range of $pH_T$ investigated, the decrease in $DMSP_T$ concentrations measured under HL conditions was, nevertheless, important (50%) over the range of $pH_T$ predicted for ocean surface waters by 2100 (8.1 to 7.6, approximately corresponding to proton concentration between 0.1 and $3.0 \times 10^{-8}$ mol $L^{-1}$ on Fig. 8a). The mean abundance of nanophytoplankton varied little within this $pH_T$ range, while the abundance of picophytoplankton increased (Fig. 5b,c). The pH-induced decrease in $DMSP_T$ under HL conditions could thus be related to a decrease in algal cellular DMSP quota. The more ambiguous response of $DMSP_T$ to the $pH_T$ decrease, compared to DMS, is in line with previous experiments that have also observed similar responses to OA for those two sulfur compounds (Vogt et al., 2008; Hopkins et al., 2010; Archer et al., 2013; Hopkins and Archer, 2014; Webb et al., 2015). Together, these results suggest that ongoing OA will have a stronger impact on the algal and bacterial DMSP transformation into DMS than on the synthesis of DMSP by algae (Vogt et al., 2008; Hopkins et al., 2010; Webb et al. 2016).

As in this study, several OA experiments conducted in mesocosms or with monospecific phytoplankton cultures have revealed a consistent decrease in DMS concentration under the influence of decreasing pH (Avgoustidi et al., 2012; Arnold et al., 2013; see Table 6 for a summary of past meso- and microcosm experiments). The percent decreases observed during our microcosm experiment over the full range of $pH_T$ investigated are consistent with the results from other studies that also found decreases ranging from -34 to -82% (Table 6). Keeping in mind the limitations of the experimental protocols, the consistency of the DMS response to pH at different latitudes and with different planktonic assemblages suggests that the potential for DMS net production during the seasonal bloom is likely to decrease in the Arctic during the next centuries. However, it is important to also keep in mind that our short-term experiment precludes any acclimation of the algae to their new environment, something that is likely to take place in response to a more gradual change in pH. In that regard, two studies have highlighted the acclimation capacity/evolutionary adaptation of the strong DMS(P) producer *Emiliana Huxleyi* to decreases in pH (Lohbeck et al., 2012; 2014). More studies are needed to fully assess how the acclimation capacity of phytoplankton will combine with short-term physiological responses to environmental stressors to shape future DMS emissions and climate.

Multiple interrelated processes interact in the global ocean to regulate DMS dynamics. Hence, several hypotheses have been proposed to explain the observed attenuation of DMS concentrations at low pH. Whereas some authors suggest a physiological response of phytoplankton under conditions of acidification (Avgoustidi et al., 2012; Hopkins and Archer, 2014), others propose a pH-induced impact on bacterial activity (Archer et al., 2013; Webb et al.,

2015) or on zooplankton grazing (Kim et al., 2010; Park et al., 2014). In our study, the absence of DMSP and DMS gross rate measurements limits our interpretation of the observed decreasing DMS trend. Results from the few previous studies where gross rate measurements were performed do not show a consistent effect of a decrease in pH on neither DMSP synthesis nor DMS consumption (Archer et al. 2013, Hopkins and Archer 2014). Despite the unavailability of

rate measurements in our study, the dominance of diatoms, an algal group lacking DMSP lyase enzymes, suggests that bacteria may have played a critical role in the observed DMS dynamics. Low pH conditions have been reported to stimulate the productivity, and hence the carbon demand, of bacteria (Maas et al., 2013; Piontek et al., 2013; Endres et al., 2014). This increase in bacterial productivity could in turn have resulted in an increase in sulfur demand leading to a

decrease in bacterial DMS yield (% of the DMSP taken up by the bacteria and cleaved into DMS). This interpretation is supported by the linear decrease in the DMS:DMSP$_d$ ratio observed over the full range of pH$_T$ investigated (Fig. 8c, Table 4), which suggests that at lower pH bacteria increasingly favoured the demethylation over the cleavage pathways of DMSP degradation. Although the decrease of this ratio could also result from an increase in

microzooplankton grazing on diatoms (Jones et al., 1998), we found no significant relationship between the micrograzers and phytoplankton, pH or DMS. Archer et al. (2013) also proposed that the pH-induced decrease in DMS observed during their experiments could partly be explained by a decrease in bacterial DMS yield related to the increase in phytoplankton biomass and net primary production at lower pH, as well as the associated increase in bacterial protein production

and sulfur demand. In contrast with their observations, however, phytoplankton biomass did not increase but decreased with pH$_T$ during our study. Nevertheless, this does not preclude the possibility of an increase in dissolved organic carbon (DOC) release by phytoplankton at low pH as previously reported (Riebesell et al., 2013). Whether the productivity and sulfur demand of DMS-consuming bacteria are also stimulated at low pH is unknown. If so, they may also have

contributed to the pH-induced decrease in DMS reported here and in other studies.

Finally, it is unlikely that grazing played a major role in the pH-associated reduction in DMS concentrations we observed. Although removing large grazers before the incubation may have affected the relative importance of microzooplankton grazing on phytoplankton during our experiment, no relationships between protist abundance and $H^+$ or DMS were found.

Modifications in micrograzing activity at low pH have been reported to either stimulate (Kim et al., 2010) or decrease net DMS production (Park et al., 2014). Park et al. (2014) attributed the reduction in grazing to a pH-induced shift in the phytoplankton species composition, with larger diatoms outcompeting smaller DMSP-rich dinoflagellates in acidified treatments, resulting in a weaker grazing pressure on the small DMSP-rich phytoplankton species and therefore less

release of DMSP and DMS. In our study, small DMSP-rich producers were not abundant and diatoms dominated the community biomass over the full range of $pH_T$ investigated. It is important to note that dinophyceae did not survive in our incubation bags, so no conclusion can be drawn as to how they could have responded to a decrease in pH and influenced DMS net production.

**4.4 Response to the contrasting light regimes**

The two experimental light treatments applied in this study had no significant effects on the temporal evolution, magnitude and taxonomic composition of the phytoplankton bloom (Table 3). These results show that both nanophytoplankton and picophytoplankton achieved optimal growth in the range of PAR used in our treatments. During a laboratory experiment, Gilstad and

Sakshaug (1990) found that 10 Arctic diatom species exhibited negligible variations in growth rate at PAR levels ranging from 50 to 500 $\mu$mol quanta m$^{-2}$ s$^{-1}$, while their growth rate increased linearly with increased PAR from 0 to 50 $\mu$mol quanta m$^{-2}$ s$^{-1}$. During our experiment, PAR ranged from 61 to 402 $\mu$mol quanta m$^{-2}$ s$^{-1}$ (Table 2), and these values fall within the tolerance range reported by Gilstad and Sakshaug (1990). Moreover, the phytoplankton community in our

incubation was initially taken at 38 m depth where PAR irradiance was as low as 7 $\mu$mol quanta m$^{-2}$ s$^{-1}$. Thus, diatoms in our experience were already very well adapted for growing at very low light levels. As they were then capable of growing as efficiently in both the HL and LL treatment, we could assess that diatoms during our experiment were capable of adapting themselves to higher irradiances very quickly. This is supported by the high and similar $F_v/F_m$ ratios recorded

during the growth phase in both light treatments (Table 5), suggesting that photosynthesis was

equally efficient at HL and LL exposure. These results suggest that phytoplankton exiting the ice pack would not necessarily experience a light shock as severe as previously noted by others (Vance et al., 2013, Galindo et al., 2016).

Among the biological variables measured, only bacteria showed a significant response to the different light treatments, with higher abundances at LL than at HL exposure (Table 3, Fig. 5d). This may reflect the known sensitivity of bacteria to UVB radiation (Herndl et al., 1993), which was absent in our LL treatment.

We found no difference in the DMSP and DMS concentrations between the two light regimes tested, as the $DMSP_T$ and DMS variations were very similar under both LL and HL conditions (Table 3). This result was unexpected considering that several processes associated with the DMS cycle are either directly (e.g., DMS photo-oxidation) or indirectly (e.g., DMS bacterial production under high UV-R) light-sensitive (Sunda et al., 2002; Slezak et al., 2007; Galí et al., 2011). Previous studies demonstrated a strong positive correlation between the solar radiation dose (SRD) received by the plankton community in the upper mixed layer and the DMS concentrations measured in surface waters (Toole et al., 2004; Vallina and Simó, 2007). Hence, our working hypothesis was that light conditions under the ice pack (low PAR and quasi-absence of UV-R) would result in lower DMS concentrations than under open water light conditions (high PAR and high UV-R). The absence of such light response may suggest that our LL treatment, as it was already light saturating for diatoms, was not different enough from our HL treatment to trigger a different photo-induced response of diatoms. Hence, the differences in PAR and UV radiation between a 5 m mixed layer depth in ice-free water and under ponded first-year ice (or a subsurface chlorophyll layer) in summer may not be sufficient to significantly affect the net production of DMSP and DMS.

## 5 Conclusion

During this study, we demonstrated that a rapid decrease in surface water pH could negatively impact the net production of algal biomass as well as DMS concentrations in Baffin Bay waters. Irrespective of the treatment, a nanophytoplankton (diatoms mostly) and picophytoplankton bloom developed within 5 to 7 days in each of our microcosms. The growth of picophytoplankton was stimulated at low $pH_T$, whereas the diatoms, which dominated the algal community in term

of biomass at all $pH_T$ levels investigated, had their abundance negatively affected by the acidification, especially at the lowest $pH_T$ level tested when compared to the controls. These results show that OA can potentially affect the magnitude of diatoms biomass in Arctic waters and enhance the already observed shift towards smaller autotrophic cells due to increased stratification (Li et al., 2009; Tremblay et al., 2012), although our results do not account for the

acclimation/evolutionary adaptation potential of natural microbial communities.

Concurrent with the response of phytoplankton to OA, the DMS dynamics were strongly affected by decreasing the $pH_T$, with a 80% reduction in average DMS concentrations between the control and the lowest $pH_T$ investigated (from $pH_T$ 8.1 to 7.2). This result adds to conclusions found in 70% of published studies that have focused on pH-DMS dynamics and showed a decrease in

DMS concentrations as pH decreases. In contrast, the pH-induced decrease in $DMSP_T$ concentration was less pronounced, as it only decreased under the HL treatment at relatively high $pH_T$ (8.1 to 7.6). The synthesis of DMSP by unicellular algae appears to be less sensitive to OA than processes responsible for its conversion into DMS, as previously hypothesized by other authors (Vogt et al., 2008; Hopkins et al., 2010; Webb et al., 2016). The lack of rate

measurements during our study precludes a definitive explanation for this trend, but a decrease in bacterial DMS yield seems to be the most probable candidate.

Our data highlighted a remarkable similarity in responses of the phytoplankton community and DMS-related processes to experimental variations in light (LL versus HL treatment). Indeed, neither the phytoplankton community nor the dimethylated sulfur compounds exhibited

significantly different signatures between the two light treatments, which were designed to simulate contrasting light conditions, experienced by a marginal ice blooms (ice-free surface mixed layer) versus under-ice blooms (irradiance under a melting pondered ice pack) or subsurface chlorophyll maxima. Although further studies are needed to fully assess the importance of light in the context of climate change in the Arctic, our results show that Arctic

diatoms may bloom under light conditions much lower than the one tested here. This apparent capacity of Arctic diatoms to grow under extremely low light conditions should be explored in future studies. As short-term impacts of OA on the DMS cycle become clearer, future studies should focus on assessing the potential susceptibility and adaptive mechanisms of microbial DMS(P) producers, processes that likely develop on a time scale closer to the natural OA rate.


**Acknowledgements.** The authors wish to thank commander Alain Gariépy, the officers, and crew of the Canadian ice-breaker NGCC Amundsen for their support during the project. We also want to thank Glenn Cooper and Kyle Simpson of the Institute of Ocean Sciences in Sidney, BC, for providing advice and the material and equipment needed for pH measurements; Isabelle Courchesne and Gabrièle Deslongchamps for the nutrients analysis, Marjolaine Blais for the flow cytometry analyses; and Jean-Bruno Nadalini for the taxonomic analyses. This study was funded by the NSERC Discovery Grant Program and Northern Research Supplement Program (M. Levasseur, M. Gosselin), as well as by the NETCARE network (funded under the NSERC Climate Change and Atmospheric Research program), ArcticNet (The Network of Centres of Excellence of Canada), and Fisheries and Oceans Canada. This is a contribution to the research programs of NETCARE, ArcticNet and Québec-Océan.

**Authors' contribution**: R. Hussherr was responsible of the elaboration of the experimental design, the sampling process, the data analysis and processing and the redaction of this manuscript. Several co-authors provided specific data included in the manuscript and all co-authors contributed to the final edition of the paper.

The authors declare that they have no conflict of interest.

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

**Table 1.** Values of $pH_T$ (total hydrogen ion scale), the associated proton concentration $[H^+]$ dissolved inorganic carbon (DIC), total alkalinity (TA), and $pCO_2$ in the microcosms under low light (LL) and high light (HL) treatments on days T0, T1, T4, and T9. Values of $pCO_2$ were calculated using $CO_2$SYS software, values of proton concentrations were calculated after the $pH_T$ measurements. 1L and 1H are the control microcosms.

| | | $pH_T$ (total hydrogen ion scale) | | | $[H^+]$ (x $10^{-8}$ mol $L^{-1}$) | | | DIC ($\mu$mol kg$^{-1}$ SW) | | | TA ($\mu$mol kg$^{-1}$ SW) | | | $pCO_2$ ($\mu$atm) | | |
|---|---|---|---|---|---|---|---|---|---|---|---|---|---|---|---|---|
| | | T0 | | | T0 | | | T0 | | | T0 | | | T0 | | |
| | | 7.94 | | | 1.15 | | | 2137 | | | 2243 | | | 492 | | |
| | | T1 | T4 | T9 | T1 | T4 | T9 | T1 | T4 | T9 | T1 | T4 | T9 | T1 | T4 | T9 |
| Microcosm | 1L | 7.94 | 7.99 | 8.22 | 1.15 | 1.03 | 0.60 | 2142 | 2129 | 2041 | 2238 | 2243 | 2247 | 509 | 451 | 247 |
| | 2L | 7.79 | 7.84 | 8.11 | 1.62 | 1.46 | 0.77 | 2186 | 2178 | 2083 | 2237 | 2243 | 2243 | 738 | 658 | 326 |
| | 3L | 7.65 | 7.70 | 8.06 | 2.24 | 1.99 | 0.87 | 2221 | 2222 | 2099 | 2231 | 2248 | 2238 | 1040 | 917 | 375 |
| | 4L | 7.46 | 7.49 | 7.83 | 3.47 | 3.24 | 1.47 | 2261 | 2255 | 2163 | 2217 | 2220 | 2226 | 1618 | 1508 | 660 |
| | 5L | 7.34 | 7.36 | 7.65 | 4.57 | 4.37 | 2.23 | 2293 | 2293 | 2210 | 2212 | 2217 | 2222 | 2116 | 2046 | 1022 |
| | 6L | 7.16 | 7.17 | 7.33 | 6.92 | 6.73 | 4.68 | 2366 | 2340 | 2283 | 2210 | 2192 | 2198 | 3296 | 3149 | 2179 |
| | 1H | 7.94 | 8.00 | 8.22 | 1.15 | 1.01 | 0.61 | 2144 | 2129 | 2041 | 2242 | 2246 | 2245 | 505 | 442 | 249 |
| | 2H | 7.77 | 7.85 | 8.12 | 1.70 | 1.42 | 0.76 | 2184 | 2169 | 2075 | 2229 | 2237 | 2237 | 768 | 638 | 322 |
| | 3H | 7.64 | 7.69 | 8.02 | 2.29 | 2.06 | 0.96 | 2218 | 2210 | 2109 | 2225 | 2231 | 2233 | 1063 | 946 | 415 |
| | 4H | 7.46 | 7.49 | 7.85 | 3.47 | 3.27 | 1.42 | 2260 | 2257 | 2157 | 2217 | 2222 | 2225 | 1610 | 1521 | 636 |
| | 5H | 7.30 | 7.32 | 7.53 | 5.01 | 4.82 | 2.92 | 2304 | 2302 | 2236 | 2208 | 2211 | 2214 | 2337 | 2259 | 1349 |
| | 6H | 7.16 | 7.17 | 7.30 | 6.92 | 6.73 | 5.00 | 2330 | 2326 | 2281 | 2178 | 2178 | 2185 | 3208 | 3134 | 2318 |

**Table 2.** Photosynthetically active radiation (PAR), ultraviolet A radiation (UVA) and ultraviolet B radiation (UVB) measured during the 9-day microcosm experiment. Values are shown for the incident irradiance and the estimated irradiance under the high light (HL) and low light (LL) treatments (i.e. values corrected for the transmission through the incubation bags). PAR values were averaged over a day whereas UVA and UVB were measured each day around noon between T5 and T9 only due to instrument dysfunction during the 4 first days.

| | | PAR ($\mu$mol quanta m$^{-2}$ s$^{-1}$) | | | UV-A (W m$^{-2}$) | | | UV-B (W m$^{-2}$) | | |
|---|---|---|---|---|---|---|---|---|---|---|
| | | Incident | HL | LL | Incident | HL | LL | Incident | HL | LL |
| Time (day) | T1 | 189 | 146 | 61 | | | | | | |
| | T2 | 216 | 168 | 70 | | | | | | |
| | T3 | 333 | 258 | 109 | | | | | | |
| | T4 | 402 | 312 | 131 | | | | | | |
| | T5 | 394 | 306 | 129 | 13 | 7.8 | 2.6 | 0.37 | 0.12 | - |
| | T6 | 381 | 270 | 124 | 4.6 | 2.8 | 0.94 | 0.2 | 0.07 | |
| | T7 | 348 | 304 | 113 | 12 | 7.6 | 2.5 | 0.37 | 0.12 | |
| | T8 | 392 | 304 | 128 | 10 | 6.2 | 2.1 | 0.36 | 0.12 | |
| | T9 | 272 | 211 | 89 | 4.9 | 3 | 1 | 0.17 | 0.06 | |

**Table 3.** Overview of the results of generalized least square models testing for the effects of time, light, and $pH_T$ over the duration of the incubation on nitrate, silicic acid, soluble reactive phosphate (SRP), chlorophyll $a$, dissolved DMSP, total DMSP, and DMS concentrations, as well as nanophytoplankton, picophytoplankton, and bacterial abundances. Type of transformations applied to data when necessary are indicated in parentheses, significant results are indicated in 1120 bold, df are the degrees of freedom, p is the significance of the t-value.

| Response variable | Factor | df | t-value | p |
|---|---|---|---|---|
| Nitrate ($\mu$mol L$^{-1}$) | Time | 77 | -7.853 | **< 0.001** |
| | Light | 77 | -0.395 | 0.693 |
| | pH | 77 | -5.156 | **< 0.001** |
| Silicic acid ($\mu$mol L$^{-1}$) | Time | 108 | -8.308 | **< 0.001** |
| | Light | 108 | 0.123 | 0.903 |
| | pH | 108 | -6.608 | **< 0.001** |
| (sqrt) SRP ($\mu$mol L$^{-1}$) | Time | 108 | -8.487 | **< 0.001** |
| | Light | 108 | -1.826 | 0.071 |
| | pH | 108 | -6.915 | **< 0.001** |
| (sqrt) Chlorophyll $a$ ($\mu$g L$^{-1}$) | Time | 108 | 7.456 | **< 0.001** |
| | Light | 108 | 0.151 | 0.880 |
| | pH | 108 | 4.862 | **< 0.001** |
| (sqrt) Nanophytoplankton (cells mL$^{-1}$) | Time | 107 | 17.614 | **< 0.001** |
| | Light | 107 | -0.346 | 0.608 |
| | pH | 107 | 6.513 | **< 0.001** |
| (sqrt) Picophytoplankton (cells mL$^{-1}$) | Time | 107 | 26.926 | **< 0.001** |
| | Light | 107 | 1.388 | 0.185 |
| | pH | 107 | 2.043 | **0.035** |

| | | | | |
|---|---|---|---|---|
| (2 outliers removed) Bacteria (cells mL$^{-1}$) | Time | 104 | 4.300 | **< 0.001** |
| | Light | 104 | 2.604 | **0.011** |
| | pH | 104 | 3.167 | **< 0.01** |
| | | | | |
| Total DMSP (nmol L$^{-1}$) | Time | 108 | 1.562 | 0.121 |
| | Light | 108 | 0.950 | 0.344 |
| | pH | 108 | 2.070 | **0.041** |
| | | | | |
| (log) Dissolved DMSP (nmol L$^{-1}$) | Time | 60 | 2.529 | **0.014** |
| | Light | 60 | -0.753 | 0.454 |
| | pH | 60 | 0.362 | 0.718 |
| | | | | |
| (sqrt) DMS (nmol L$^{-1}$) | Time | 77 | 2.478 | **0.015** |
| | Light | 77 | 0.387 | 0.700 |
| | pH | 77 | 6.635 | **< 0.001** |

**Table 4.** Empirical relationships of various biological or chemical (response) variables with proton concentration (mol $L^{-1}$). Values are the means of all the microcosms. Linear regressions were performed on both light treatments when light had significant effect on the response variable (see Table 3). Significant results are in bold, df are the degrees of liberty, $r^2$ is the squared Pearson correlation coefficient between proton concentration and the response variable, and p is the significance of each parameters of the regression.

| Response variable | df | $r^2$ | Parameters | | p |
|---|---|---|---|---|---|
| Mean Chlorophyll *a* ($\mu$g $L^{-1}$) | 10 | 0.86 | slope | $-2.70\ 10^7$ | **< 0.001** |
| | | | intercept | 3.95 | **< 0.001** |
| Mean nanophytoplankton (cells $mL^{-1}$) | 10 | 0.73 | slope | $-4.37\ 10^{10}$ | **< 0.001** |
| | | | intercept | $6.74\ 10^3$ | **< 0.001** |
| Mean picophytoplankton (cells $mL^{-1}$) | 10 | 0.32 | slope | $6.52\ 10^{11}$ | **0.056** |
| | | | intercept | $6.73\ 10^4$ | **< 0.001** |
| Mean bacteria (cells $mL^{-1}$) | | | | | |
| *High Light* | 4 | 0.84 | slope | $-2.54\ 10^{12}$ | **< 0.01** |
| | | | intercept | $9.77\ 10^5$ | **< 0.001** |
| *Low Light* | 4 | 0.33 | slope | $-8.04\ 10^{11}$ | 0.23 |
| | | | intercept | $1.01\ 10^6$ | **< 0.001** |
| Mean DMSP$_T$ (nmol $L^{-1}$) | 10 | 0.04 | slope | $-6.90\ 10^7$ | 0.55 |
| | | | intercept | $3.81\ 10^1$ | **< 0.001** |
| Mean DMS (nmol L-1) | 10 | 0.9 | slope | $-1.19\ 10^8$ | **< 0.001** |
| | | | intercept | 8.37 | **< 0.001** |
| Mean DMS:DMSPd ratio | 10 | 0.57 | slope | $-1.02\ 10^8$ | **< 0.01** |
| | | | intercept | 8.45 | **< 0.001** |

**Table 5.** $F_v/F_m$ ratios for three different $pH_T$ measured on days T2, T4, T6, T7 and T9 under high light (HL) and low light (LL) conditions.

| | pH | 8.07 | | 7.81 | | 7.22 | |
|---|---|---|---|---|---|---|---|
| | | HL | LL | HL | LL | HL | LL |
| Time (day) | T2 | 0.52 | 0.55 | 0.51 | 0.53 | 0.44 | 0.46 |
| | T4 | 0.59 | 0.58 | 0.58 | 0.59 | 0.5 | 0.5 |
| | T6 | 0.49 | 0.47 | 0.44 | 0.51 | 0.5 | 0.5 |
| | T7 | 0.4 | 0.4 | 0.41 | 0.42 | 0.5 | 0.52 |
| | T9 | 0.31 | 0.32 | 0.33 | 0.36 | 0.36 | 0.37 |


**Table 6.** Changes in dimethylsulfide (DMS) concentrations reported in previous $CO_2$ perturbation experiments in mesocosms or microcosms. $\Delta$pH represents the change in $pH_T$ applied in each experiment. In our study, $pH_T$ ranged from 7.94 to 7.16 at day T1. ND: No data

| Location | $\Delta$ pH | $pCO_2$ range ($\mu$atm) | Change in DMS (%) | Reference |
|---|---|---|---|---|
| Baffin Bay, Arctic | -0.75 | 500 - 3000 | -80 | This study |
| Baltic Sea | -0.4 | 350 - 1500 | -34 | Webb et al. 2016 |
| Raunesfjorden, Norway | -0.6 | 280 - 3000 | -60 | Webb et al. 2015 |
| Jangmok, Korea | -0.5 | 160 - 830 | -82 | Park et al. 2014 |
| Kongsfjorden, Svalbard | -0.8 | 180 - 1420 | -60 | Archer et al. 2013 |
| Raunesfjorden, Norway | -0.2 | 300 - 750 | -40 | Avgoustidi et al. 2013 |
| Raunesfjorden, Norway | -0.3 | 300 - 750 | -57 | Hopkins et al. 2010 |
| Raunesfjorden, Norway | -0.5 | 300 - 750 | 0 | Vogt et al. 2008 |
| NW European Shelf | -0.4 | 340 - 1000 | 225 | Hopkins et Archer 2014 |
| Jangmok, Korea | ND | 400 - 900 | 80 | Kim et al. 2010 |


# Figures

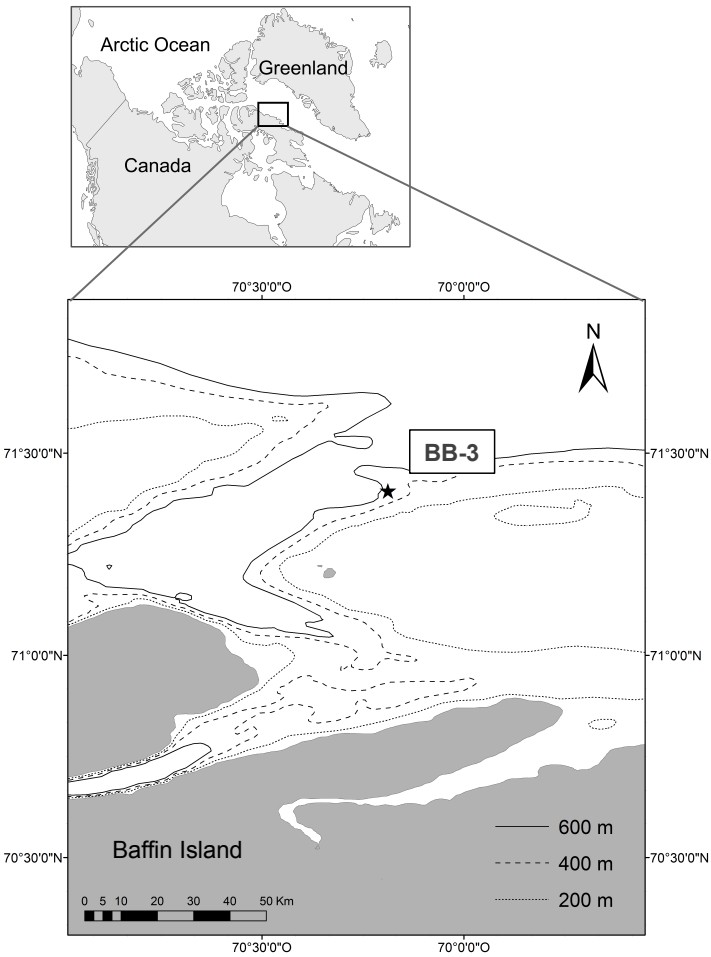

**Fig. 1.** Map showing the location of station BB-3 (71°24.373'N; 70°11.269'W) where seawater was collected for the incubation experiment.


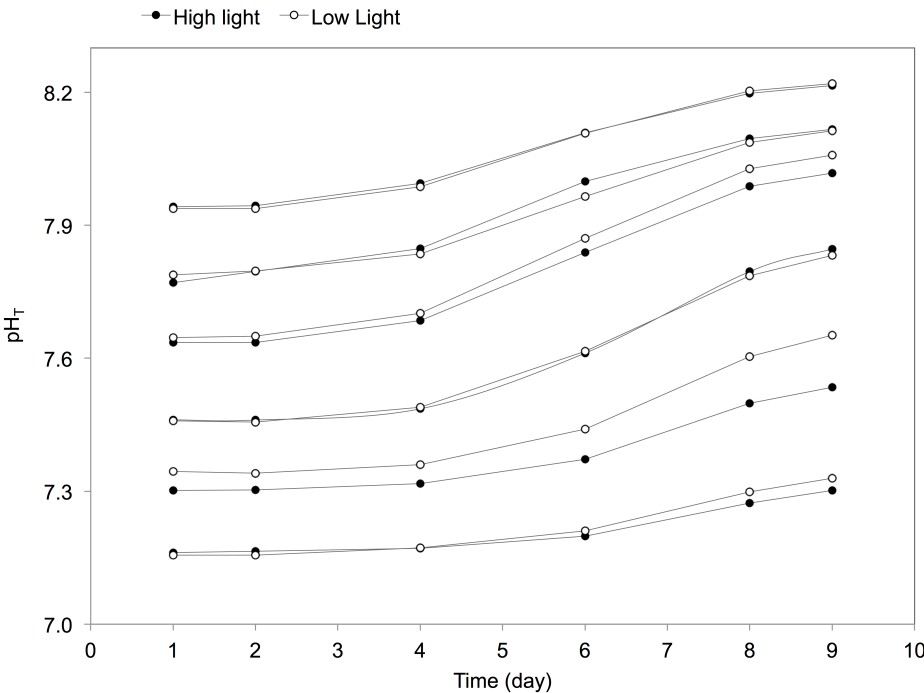

**Fig. 2.** Temporal variations in $pH_T$ (total hydrogen ion scale) during the microcosm experiment. Black and white circles represent the pH gradient in the high light (HL) and low light (LL) treatments, respectively.


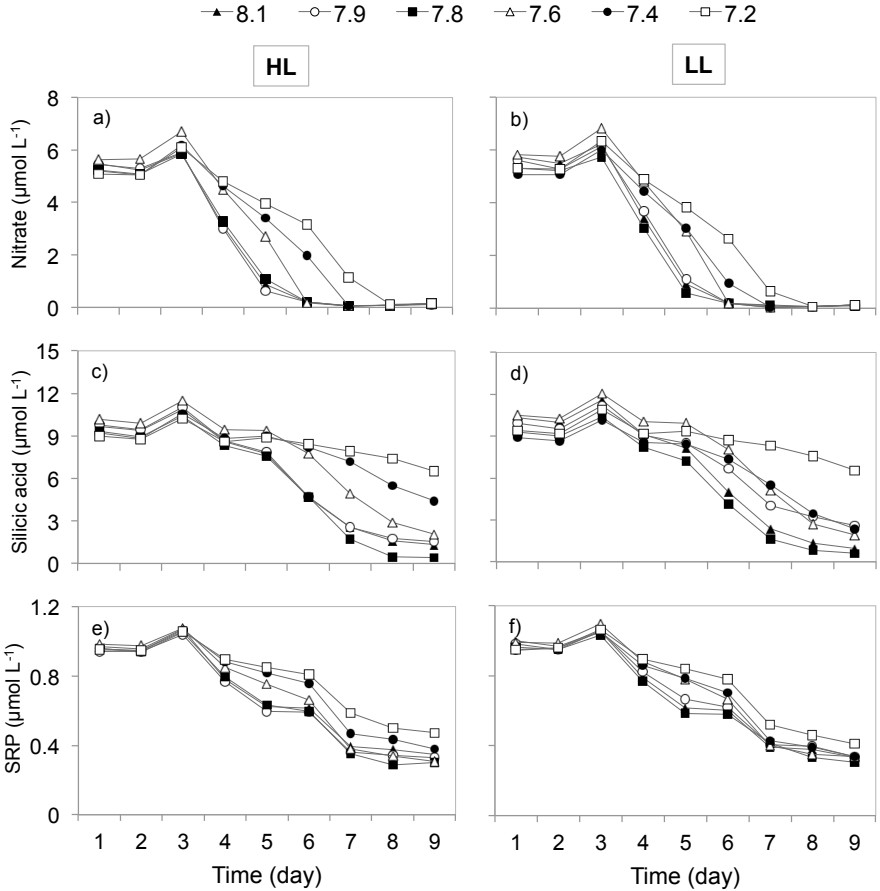

**Fig. 3.** Temporal variations in (a,b) nitrate, (c,d) silicic acid, and (e,f) soluble reactive phosphate (SRP) concentrations during the experiment. Left and right panels show variations under the high light (HL) and low light (LL) treatments, respectively. Each curve represents a microcosm identified by its mean $pH_T$ value.

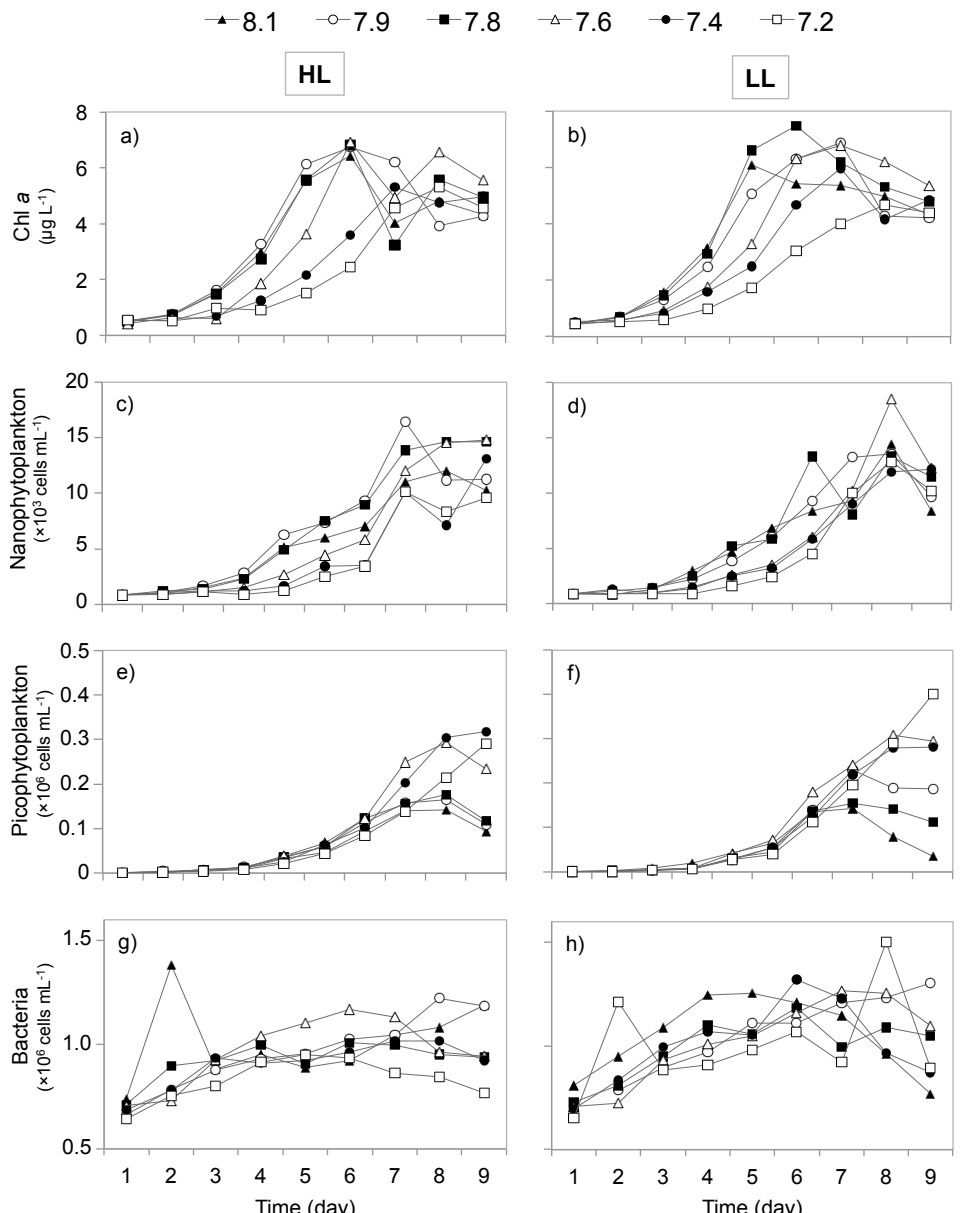

**Fig. 4.** Temporal variations in (a,b) chlorophyll *a* (Chl *a*) concentration, (c,d) nanophytoplankton, (e,f) picophytoplankton, and (g,h) bacteria during the microcosm experiment. Left and right panels show variations under the high light (HL) and low light (LL) treatments, respectively. Each curve represents a microcosm identified by its mean $pH_T$ value.

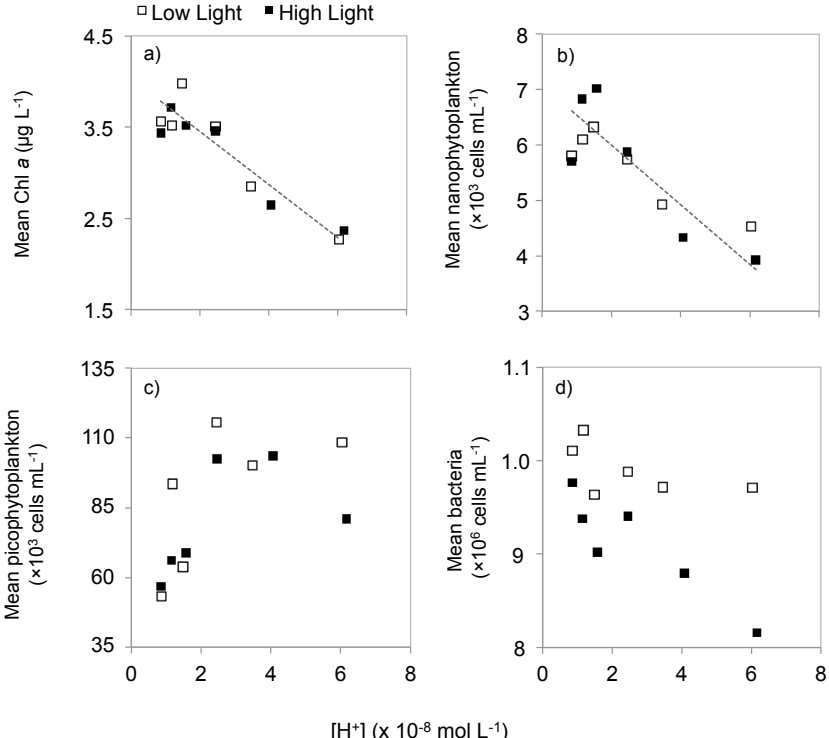

**Fig. 5.** Relationships between the mean proton concentration values ([H$^+$]) and a) mean Chlorophyll *a* (Chl *a*), b) mean nanophytoplankton abundance, c) mean picophytoplankton abundance, and d) mean bacteria abundance. Values are the means over the duration of the incubation in each microcosm. Significant regressions are shown with dashed lines; details of which are given in Table 4. Regressions were based on the full dataset when no significant

difference between the light treatments was detected (see also Table 3 for more details).

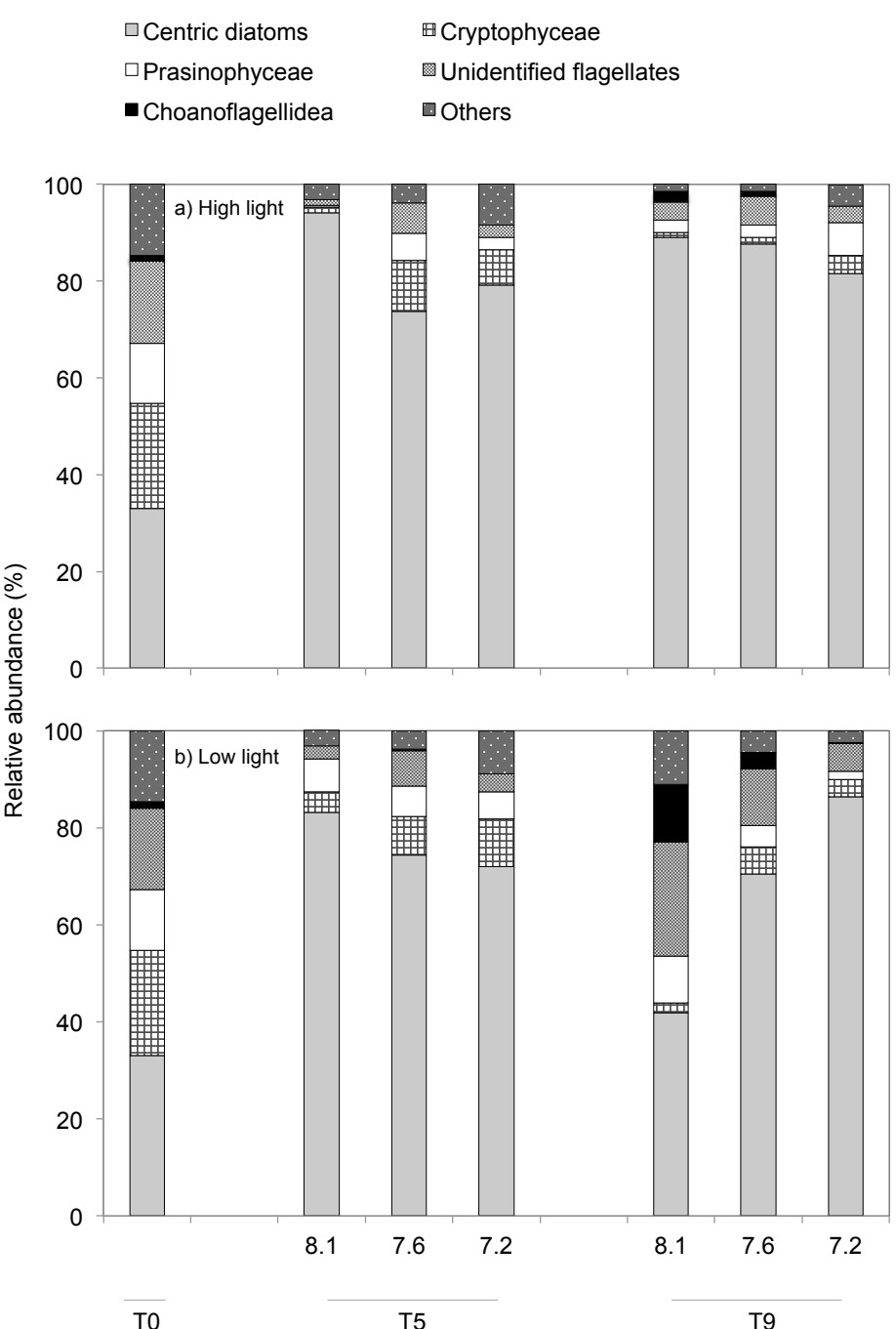

**Fig. 6.** Relative abundance of 6 groups of protists at the beginning (T0), the middle (T5) and the end (T9) of the microcosm experiment for (a high light (HL) and (b low light (LL) treatments. The group "Others" includes pennate diatoms, dinophyceae, chrysophyceae, dictyophyceae, euglenophyceae, ciliates and unidentified cells. Each barplot represents a pH treatment. The barplot at T0 represents the initial community assemblage before pH manipulations, and is therefore the same for both light treatments.



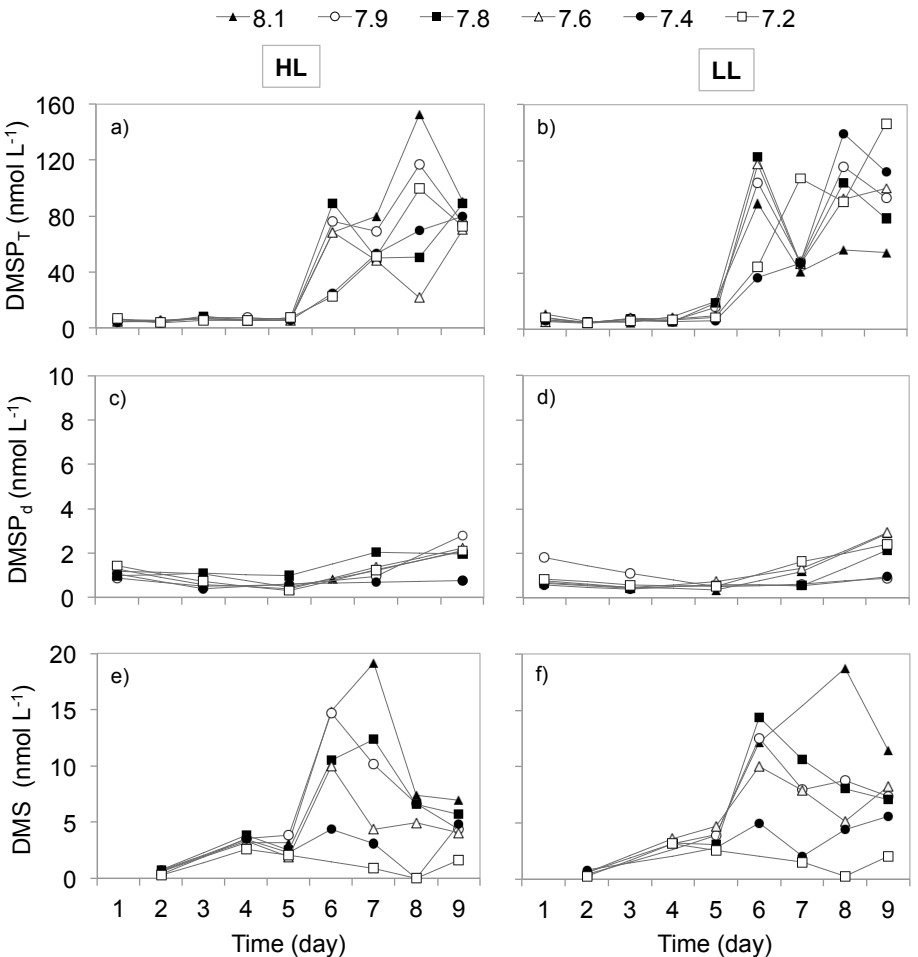

**Fig. 7.** Temporal variations in (a,b) total DMSP (DMSP$_T$), (c,d) dissolved DMSP (DMSP$_d$), and (e,f) DMS concentrations during the microcosm experiment. Left and right panels show variations under the high light (HL) and low light (LL) treatments, respectively. Each curve represents a microcosm identified by its mean pH$_T$ value.


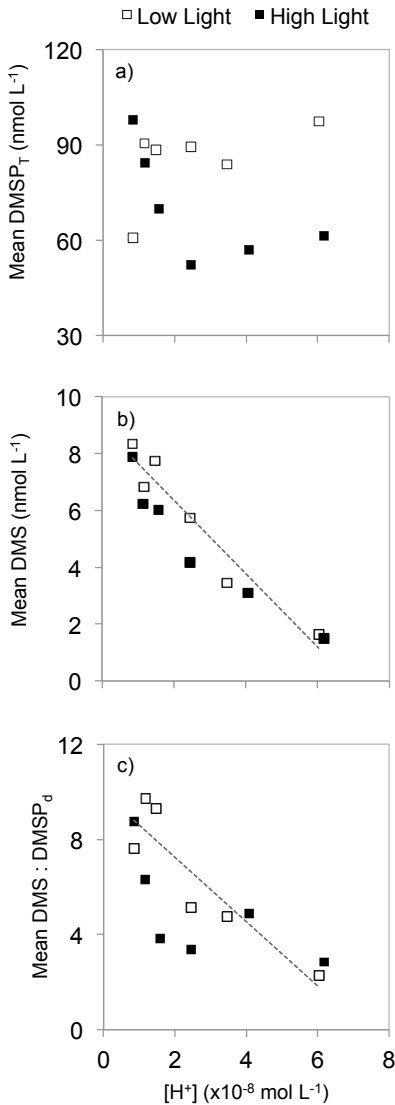

**Fig. 8.** Relationships between the mean proton concentration values ($[H^+]$) and (a) mean total DMSP concentration, (b) mean DMS concentration, and (c) mean DMS:DMSP$_d$ ratio. Values are the mean over the duration of the incubation in each microcosm. Significant regressions are shown as dashed lines, details of which are given in Table 4.