# Peer review of "Impact of ocean acidification on Arctic phytoplankton blooms and dimethylsulfide concentration under simulated ice-free and under-ice conditions"

_Biogeosciences, 2016_

## Referee Comment (RC1) · Anonymous Referee #1 · 9 Jan 2017

Summary

Hussherr et al. present an interesting and timely study that addresses the lack of data we have on the response of DMS concentrations in Arctic waters to ocean acidification. Specifically, the paper presents the results of a 9 day experiment in which seawater was incubated in 10 L gas tight bags under a range of pH/pCO2 treatments, from pHT 7.9 – pHT 7.2, representing a range from 'present day' to end of century to extreme far future values. Furthermore, the authors investigated the role of light, dividing the bags into low light and high light treatments, in order to simulate ice free and under ice

conditions. The pH gradient method is an established and well-used technique, most useful when the possibility of replication is limited. Acidification was performed using the addition of strong acid and base, again another established technique. Samples for a range of parameters were taken on a regular basis over the 9 day experiment. Within 3 days of the start of the incubation period, a bloom initiated in all bags, leading to an increase in phytoplankton biomass and DMS/DMSP concentrations – differences in the response were attributed to the pH treatments, with no clear observed effect of light. DMS concentrations significantly decreased with decreasing pH, which is in agreement with the one other previous study from Arctic waters (Archer et al. 2013), leading to the conclusion that DMS concentrations during Arctic blooms may be lower in the future, with possible implications for the Arctic climate. The paper is generally well written and logically structured. I have identified a number of minor issues that the authors should address, relating to the methods and the bloom dynamics. Assuming the authors make the suggested changes, this paper would be suitable for publication in Biogeosciences.

Key points

1. Methods: L128: the authors state they 'poured' seawater into the gas-tight bags. Through a luer valve? Some clarity is needed as to their exact methods. Pouring is not recommended when handling gas sensitive samples as the gas phase equilibrium may be altered. Notwithstanding the difficulty in pouring anything through a luer valve! Some more detailed explanation is required.

L131: samples were incubated at 4.3 ± 1.6 °C. This seems warm for experiments that are attempting to simulate 'under ice' conditions. Can the authors provide some justification/ further explanation?

2. What stimulated the bloom in the bags? Were the team expecting a bloom to occur in the way it did in the bags? Did a bloom also develop in the sampled water simultaneously (i.e. was this a natural or artificial bloom?)? Many questions. . .therefore

some more discussion would be useful to the reader. After all, without such a nice bloom, it is unlikely a DMS(P) response would have been observed. L492: the authors talk about their findings in the context of the Arctic spring phytoplankton bloom – but actually this experiment sampled waters in August, which must qualify as late summer for the Arctic. So how comparable were the starting conditions to the spring bloom?

3. Reference to Richier et al. (2014) (Phytoplankton responses and associated carbon cycling during shipboard carbonate chemistry manipulation experiments conducted around Northwest European shelf seas) is lacking and should be included in the discussions. The work of Richier et al. is the most similar to this study in terms of the experimental techniques used. The authors do cite Hopkins & Archer (2014) which was part of the same study, but only in a DMS(P) context. The shipboard incubations of Richier et al. and Hopkins & Archer also need to be addressed in the context of this study in terms of the phytoplankton response.

Specific comments and suggestions

Title: it would be more accurate to say 'DMS concentrations', as 'production' implies that the work include rate measurements.

L45 – 49: These two sentences are somewhat ambiguous and need further explanation. Why is climate change 'faster and more important' in the Arctic? In what respect?

L50 – 52: this sentence seems detached and slightly out of context. I see what the authors intend by it. Perhaps they could re-phrase so it says something like: 'Given that the reduction in extent and thickness of sea ice cover and the acidification of surface waters can potentially impact primary productivity, it is important to consider the associated effects on the production of biogenic climate-active gases. . .' or similar, just to change the emphasis slightly, and provide an impetus for the work.

L70 – 75, and throughout: the authors make no mention of Richier et al (2014), a recent and relevant paper that should be cited.

L80: re-word. Suggest: 'Emissions of DMS thus can. . .'

L83: add 'atmosphere' at end of sentence (so reads 'summer Arctic atmosphere')

L97: Not necessary to cite Webb at this point as it is not a review paper. Fine to just cite the references as you specifically mention them later in the paragraph.

L99: although Archer et al. is mentioned later in the paragraph in an Arctic specific context, it would be appropriate to add it to the listed references here.

L142: 'submitted' would be better substituted for 'exposed'

L397: should read 'species'

L403: to improve readability, re-phrase: 'The sole exception was the LL control mesocosm. . .'

L452: Rather than staying 'high pHT', it would be useful to state the range of pH over which the response was observed.

L513 – 517: this long sentence needs some re-wording as it is currently hard to follow and the English is poor in places.

L524: should read 'switched'

L527 – 528: needs re-wording. Suggest: 'These results also suggest that diatoms could have more difficulty in efficiently taking up/assimilating. . .'

Section 4.2: some discussion of the results in comparison to the findings of Richier et al. would be useful, as the two studies use very similar techniques – yet yield quite contrasting responses.

L608 – 610: Archer et al (2013) and Hopkins and Archer (2014) report rate measurements – so this statement is not correct, and their findings should be included in the discussion.

---

## Referee Comment (RC2) · Anonymous Referee #2 · 22 Jan 2017

The manuscript provides a good account of the potential effects on OA of Baffin Bay seawater in the Arctic Ocean and it's affect on various variables such as Chl a, pH, nutrients, DMSPt and DMS etc., The manuscript is well presented and figures and tables are very clearly produced. Significant changes have been highlighted in the 10 day incubation experiment. Whilst the authors state that the rapid change in pH investigated over 10 days is not representative of the gradual OA that is taking place their study does reflect potential extreme resonses. However, some further acknowledgement of this should be made in the discussion and in particular acknowledge that organisms do adapt to changes which may well affect the validity of some the of discussion and

conclusions.

The abstract should contain more of the important findings mentioned in the text. Go through and highlight these changes in discussion and make sure they are included in the abstract.

The introduction is well stated although there should be some attempt perhaps in the discussion to state why different authors find different affects of OA on phytoplankton response.

Methods. Are the expts 9 days or 10 days-it is not clear. As the authors removed the large grazers could microzooplankton affected the results? Why was alkalinity kept constant? Surely in the natural environment and in particular a bloom event alkalinity would change as well as the concentration and ion activities of some of the constituents measured?

Results: See the sticky notes added to the manuscript and please attend to them. Can you say what species were mainly reflected in the nannoplankton. Were any calcareous?

Discussion and Conclusion: see the sticky notes. These parts need to be carefully gone over and some sentences modified.

Overall I would recommend publication with attention paid to the minor comments. Also the authors should end their discussion with what future studies should concentrate on wrt. Baffin Bay to extend the field and make these expts more relevant to actual conditions in the field..

Please also note the supplement to this comment:
http://www.biogeosciences-discuss.net/bg-2016-501/bg-2016-501-RC2-supplement.pdf

**Supplement:**

**Impact of ocean acidification on Arctic phytoplankton blooms and dimethylsulfide production under simulated ice-free and under-ice conditions**

5   **Hussherr, Rachel[1]; Levasseur, Maurice[1]; Lizotte, Martine[1]; Tremblay, Jean-Éric[1]; Mol, Jacoba[2]; Thomas, Helmuth[2]; Gosselin, Michel[3]; Starr, Michel[4]; Miller, Lisa Ann[5]; Jarniková, Tereza[6]; Schuback, Nina[6]; Mucci, Alfonso[7]**

[1] Québec-Océan and Takuvik joint UL-CNRS laboratory, Département de biologie, Université Laval, Québec, Québec G1V 0A6, Canada

10  [2] Department of Oceanography, Dalhousie University, Halifax, Nova Scotia B3H 4R2, Canada

[3] Institut des sciences de la mer de Rimouski, Université du Québec à Rimouski, Rimouski, Québec G5L 3A1, Canada

[4] Maurice Lamontagne Institute, Fisheries and Oceans Canada, Mont-Joli, Québec G5H 3Z4, Canada

15  [5] Institute of Ocean Sciences, Fisheries and Oceans Canada, Sidney, British Columbia V8L 4B2, Canada

[6] Department of Earth, Ocean and Atmospheric Sciences, University of British Columbia, Vancouver, British Columbia V6T 1Z4, Canada

[7] GEOTOP and Department of Earth and Planetary Sciences, McGill University, Montréal,
20  Québec H3A 0E8, Canada

*Correspondence to*: Rachel Hussherr (rachel.hussherr@gmail.com)

**Abstract.** In an experimental assessment of the potential impact of Arctic Ocean acidification on seasonal phytoplankton blooms and associated dimethylsulfide (DMS) dynamics, we incubated water from Baffin Bay under conditions representing an acidified Arctic Ocean. Using two light regimes simulating under-ice/ subsurface chlorophyll maxima (low light; Low PAR and no UVB) and ice-free (high light; High PAR + UVA + UVB) conditions, water collected at 38 m was exposed over 9 days to 6 levels of decreasing pH from 8.1 to 7.2. A phytoplankton bloom dominated by the centric diatoms *Chaetoceros* spp. reaching up to 7.5 $\mu$g chlorophyll $a$ L$^{-1}$ took place in all experimental bags. Total dimethylsulfoniopropionate (DMSP$_T$) and DMS concentrations reached 155 nmol L$^{-1}$ and 19 nmol L$^{-1}$, respectively. Under both light regimes, chlorophyll $a$ and DMS concentrations decreased linearly with increasing proton concentration at all pH tested. Concentrations of DMSP$_T$ also decreased but only under high light and over a smaller pH range (from 8.1 to 7.6). In contrast to nanophytoplankton (2-20 $\mu$m), picophytoplankton ($\leq 2\,\mu$m) was stimulated by the decreasing pH. We furthermore observed no significant difference between the two light regimes tested in term of chlorophyll $a$, phytoplankton abundance/ taxonomy, and DMSP/ DMS net concentrations. These results show that OA could significantly decrease the algal biomass and inhibit DMS production during the seasonal phytoplankton bloom in the Arctic, with possible consequences for the regional climate.

**1 Introduction**

As a result of anthropogenic emissions of carbon dioxide ($CO_2$) to the atmosphere, important transformations are observed in the global ocean, including a rise in water temperature, a decrease in ocean pH, modifications of water circulation patterns and nutrient distributions, and a loss of sea-ice in the Arctic (ACIA, 2005; Fabry et al., 2009; Macdonald et al., 2015). Climate change is faster and more important in the Arctic than in any other place in the world, 
[revised manuscript text omitted]

Multiple interrelated processes interact in the global ocean to regulate DMS dynamics. Hence, several hypotheses have been proposed to explain the observed attenuation of DMS concentrations at low pH. Whereas some authors suggest a physiological response of phytoplankton under conditions of acidification (Avgoustidi et al., 2012; Hopkins and Archer, 2014), others propose a pH-induced impact on bacterial activity (Archer et al., 2013; Webb et al., 2015) or on zooplankton grazing (Kim et al., 2010; Park et al., 2014). Unfortunately, these hypotheses remain to be confirmed, given the lack of DMS production and degradation rate measurements. The absence of DMSP and DMS rate measurements in our study also limits our interpretation of the observed decreasing DMS trend, but the dominance of diatoms and the absence of DMSP lyase in that algal group suggest that bacteria may have played a critical role. Low pH conditions have been reported to stimulate the productivity, and hence the carbon demand, of bacteria (Maas et al., 2013; Piontek et al., 2013; Endres et al., 2014). This increase in bacterial productivity could in turn have resulted in an increase in sulfur demand leading to a decrease in bacterial DMS yield (% of the DMSP taken up by the bacteria and cleaved into DMS). This interpretation is supported by the linear decrease in the DMS:DMSP$_d$ ratio observed

over the full range of $pH_T$ investigated (Fig. 8c, Table 4), which suggests that at lower pH bacteria increasingly favoured the demethylation over the cleavage pathways of DMSP degradation. Archer et al. (2013) also proposed that the pH-induced decrease in DMS observed during their experiments could partly be explained by a decrease in bacterial DMS yield related to the increase in phytoplankton biomass and net primary production at lower pH, as well as the associated increase in bacterial protein production and sulfur demand. In contrast with their observations, however, phytoplankton biomass did not increase but decreased with $pH_T$ during our study. Nevertheless, this does not preclude the possibility of an increase in dissolved organic carbon (DOC) release by phytoplankton at low pH as previously reported (Riebesell et al., 2013). Whether the productivity and sulfur demand of DMS-consuming bacteria are also stimulated at low pH is unknown. If so, they may also have contributed to the pH-induced decrease in DMS reported here and in other studies.

Finally, it is unlikely that grazing played a major role in the pH-associated reduction in DMS concentrations we observed. Modifications in micrograzing activity at low pH have been reported to either stimulate (Kim et al., 2010) or decrease net DMS production (Park et al., 2014). Park et al. (2014) attributed the reduction in grazing to a pH-induced shift in the phytoplankton species composition, with larger diatoms outcompeting smaller DMSP-rich dinoflagellates in acidified treatments, resulting in a weaker grazing pressure on the small DMSP-rich phytoplankton species and therefore less release of DMSP and DMS. In our study, small DMSP-rich producers were not abundant and diatoms dominated the community biomass over the full range of $pH_T$ investigated. It is important to note that dinophyceae did not survive in our incubation bags, so no conclusion can be drawn as to how they could have responded to a decrease in pH and influenced DMS net production.

**4.4 Response to the contrasting light regimes**

The two experimental light treatments applied in this study had no significant effects on the temporal evolution, magnitude and taxonomic composition of the phytoplankton bloom (Table 3). These results show that both nanophytoplankton and picophytoplankton achieved optimal growth in the range of PAR used in our treatments. During a laboratory experiment, Gilstad and Sakshaug (1990) found that 10 Arctic diatom species exhibited negligible variations in growth rate at PAR levels ranging from 50 to 500 $\mu$mol quanta m$^{-2}$ s$^{-1}$, while their growth rate increased

linearly with increased PAR from 0 to 50 $\mu$mol quanta m$^{-2}$ s$^{-1}$. During our experiment, PAR ranged from 61 to 402 $\mu$mol quanta m$^{-2}$ s$^{-1}$ (Table 2), and these values fall within the tolerance range reported by Gilstad and Sakshaug (1990). Moreover, the phytoplankton community in our incubation was initially taken at 38 m depth were PAR irradiance was as low as 7 $\mu$mol quanta m$^{-2}$ s$^{-1}$. Thus, diatoms in our experience were already very well adapted for growing at very low light le[.] As they were then capable of growing as efficiently in both the HL and LL treatment, we could assess that diatoms during our experiment were capable of adapting themselves to higher irradiances very quickly. This is supported by the high and similar $F_v/F_m$ ratios recorded during the growth phase in both light treatments (Table 5), suggesting that photosynthesis was equally efficient at HL and LL exposure.

Among the biological variables measured, only bacteria showed a significant response to the different light treatments, with higher abundances at LL than at HL exposure (Table 3, Fig. 5d). This may reflect the known sensitivity of bacteria to UVB radiation (Herndl et al., 1993), which was absent in our LL treatment.

We found no difference in the DMSP and DMS concentrations between the two light regimes tested, as the DMSP$_T$ and DMS variations were very similar under both LL and HL conditions (Table 3). This result was unexpected considering that several processes associated with the DMS cycle are either directly (e.g., DMS photo-oxidation) or indirectly (e.g., DMS bacterial production under high UV-R) light-sensitive (Sunda et al., 2002; Slezak et al., 2007; Galí et al., 2011). Previous studies demonstrated a strong positive correlation between the solar radiation dose (SRD) received by the plankton community in the upper mixed layer and the DMS concentrations measured in surface waters (Toole et al., 2004; Vallina and Simó, 2007). Hence, our working hypothesis was that light conditions under the ice pack (low PAR and quasi-absence of UV-R) would result in lower DMS concentrations than under open water light conditions (high PAR and high UV-R). The absence of such light response may suggest that our LL treatment, as it was already light saturating for diatoms, was not different enough from our HL treatment to trigger a different photo-induced response of diatoms. Hence, the differences in PAR and UV radiation between a 5 m mixed layer depth in ice-free water and under ponded first-year ice (or a subsurface chlorophyll layer) in summer may not be sufficient to significantly affect the net production of DMSP and DMS. These results furthermore suggest that SRD may not be the main

factor driving net DMS production in Arctic waters, similar to results from the northeast Atlantic, where Belviso and Caniaux (2009) found that the SRD accounted for only 19% to 24% of DMS variations during the summer.

**5 Conclusion**

During this study, we demonstrated that OA could negatively impact the net production of algal biomass as well as DMS concentrations in Baffin Bay waters. Irrespective of the treatment, a nanophytoplankton (diatoms mostly) and picophytoplankton bloom developed within 5 to 7 days in each of our microcosms. The growth of picophytoplankton was stimulated at low $pH_T$, whereas the diatoms, which dominated the algal community in term of biomass at all $pH_T$ levels investigated, had their abundance negatively affected by the acidification, especially at the lowest $pH_T$ level tested, when compared to the controls. These results show that OA can potentially affect the magnitude of diatom bloom biomass in Arctic waters and enhance the already observed shift towards smaller autotrophic cells due to increased stratification (Li et al., 1999; Tremblay et al., 2012).

Concurrent with the response of phytoplankton to OA, the DMS dynamics were strongly affected by decreasing the $pH_T$, with a 80% reduction in average DMS concentrations between the control and the lowest $pH_T$ investigated (from $pH_T$ 8.1 to 7.2). This result adds to conclusions found in 70% of published studies that have focused on pH-DMS dynamics and showed a decrease in DMS concentrations as pH decreases. In contrast, the pH-induced decrease in $DMSP_T$ concentration was less pronounced, as it only decreased under the HL treatment at relatively high $pH_T$ (8.1 to 7.6). The synthesis of DMSP by unicellular algae appears to be less sensitive to OA than processes responsible for its conversion into DMS, as previously hypothesized by other authors (Vogt et al., 2008; Hopkins et al., 2010; Webb et al., 2016). The lack of rate measurements during our study precludes a definitive explanation for this trend, but a decrease in bacterial DMS yield seems to be the most probable candidate.

Our data highlighted a remarkable similarity in responses of the phytoplankton community and DMS-related processes to experimental variations in light (LL versus HL treatment). Indeed, neither the phytoplankton community nor the dimethylated sulfur compounds exhibited significantly different signatures between the two light treatments, which were designed to

simulate contrasting light conditions, experienced by a marginal ice blooms (ice-free surface mixed layer) versus under-ice blooms (irradiance under a melting pondered ice pack) or subsurface chlorophyll maxima. Although further studies are needed to fully assess the importance of light in the context of climate change in the Arctic, these results suggest that the presence/absence of sea ice will not directly affect the instantaneous impacts of OA on phytoplankton blooms and DMS production in the Arctic Ocean.

710

**Acknowledgements.** The authors wish to thank commander Alain Gariépy, the officers, and
715     crew of the Canadian ice-breaker NGCC Amundsen for their support during the project. We also
want to thank Glenn Cooper and Kyle Simpson of the Institute of Ocean Sciences in Sidney, BC,
for providing advice and the material and equipment needed for pH measurements; Isabelle
Courchesne and Gabrièle Deslongchamps for the nutrients analysis, Marjolaine Blais for the flow
cytometry analyses; and Jean-Bruno Nadalini for the taxonomic analyses. This study was funded
720     by the NSERC Discovery Grant Program and Northern Research Supplement Program (M.
Levasseur, M. Gosselin), as well as by the NETCARE network (funded under the NSERC
Climate Change and Atmospheric Research program), ArcticNet (The Network of Centres of
Excellence of Canada), and Fisheries and Oceans Canada. This is a contribution to the research
programs of NETCARE, ArcticNet and Québec-Océan.

725

**Authors contribution**: R. Hussherr was responsible of the elaboration of the experimental
design, the sampling process, the data analysis and processing and the redaction of this
manuscript. Several co-authors provided specific data included in the manuscript and all co-
authors contributed to the final edition of the paper.

730

The authors declare that they have no conflict of interest.

[revised manuscript text omitted]

---

## Author Comment (AC1) · 2 Mar 2017

Summary

Hussherr et al. present an interesting and timely study that addresses the lack of data we have on the response of DMS concentrations in Arctic waters to ocean acidification. Specifically, the paper presents the results of a 9 day experiment in which seawater was incubated in 10 L gas tight bags under a range of pH/pCO2 treatments, from pHT 7.9 – pHT 7.2, representing a range from 'present day' to end of century to extreme far future values. Furthermore, the authors investigated the role of light, dividing the bags into low light and high light treatments, in order to simulate ice free and under ice conditions. The pH gradient method is an established and well-used technique, most useful when the possibility of replication is limited. Acidification was performed using the addition of strong acid and base, again another established technique. Samples for a range of parameters were taken on a regular basis over the 9 day experiment. Within 3 days of the start of the incubation period, a bloom initiated in all bags, leading to an increase in phytoplankton biomass and DMS/DMSP concentrations – differences in the response were attributed to the pH treatments, with no clear observed effect of light. DMS concentrations significantly decreased with decreasing pH, which is in agreement with the one other previous study from Arctic waters (Archer et al. 2013), leading to the conclusion that DMS concentrations during Arctic blooms may be lower in the future, with possible implications for the Arctic climate. The paper is generally well written and logically structured. I have identified a number of minor issues that the authors should address, relating to the methods and the bloom dynamics. Assuming the authors make the suggested changes, this paper would be suitable for publication in Biogeosciences.

In the first place, the authors wish to thank the referee for the positive and constructive comments. Every comment made has been carefully considered and the manuscript has been amended/modified when necessary by addressing each point one-by-one.

Key points

1. Methods: L128: the authors state they 'poured' seawater into the gas-tight bags. Through a luer valve? Some clarity is needed as to their exact methods. Pouring is not recommended when handling gas sensitive samples as the gas phase equilibrium may be altered. Notwithstanding the difficulty in pouring anything through a luer valve! Some more detailed explanation is required.

The water was transferred by gravity from the Niskin bottle to the gas-tight bags using a Teflon

tube connected between the output valve of the Niskin bottle and the luer-valve of the bag.

This information has been added to the text.

L137 - New sentence: "After initial collection, water was gravity filtered through a 200 $\mu$m Nitex mesh, in order to remove large grazers, and transferred to 12 gas-tight 10-L bags (HyClone Labtainer©, Thermo Scientific) using a Teflon tube linking the output valve of the Niskin bottle and the luer-valve of the bag."

L131: Samples were incubated at 4.3 ± 1.6 ℃. This seems warm for experiments that are attempting to simulate 'under ice' conditions. Can the authors provide some justification/ further explanation?

This is a good point. We are aware that the experimental temperature did not perfectly reproduce the conditions prevailing under the ice. Our deck incubator was constantly flushed with surface water during the cruise and we had no control over the temperature of the water. However, all HL and LL bags were in the same incubator, hence submitted to the same temperature.

The following sentence was added to address this issue:

L142 – 145: "Since our deck incubator was cooled with circulating surface water, we had no control on the temperature during the incubation (mean temperature of 4.3 ± 1.6℃ over the 9-day experiment). However, all bags were in the same incubator, hence submitted to the same temperature."

2. What stimulated the bloom in the bags? Were the team expecting a bloom to occur in the way it did in the bags? Did a bloom also develop in the sampled water simultaneously (i.e. was this a natural or artificial bloom?)? Many questions. . .therefore some more discussion would be useful to the reader. After all, without such a nice bloom, it is unlikely a DMS(P) response would have been observed. L492: the authors talk about their findings in the context of the Arctic spring phytoplankton bloom – but actually this experiment sampled waters in August, which must qualify as late summer for the Arctic. So how comparable were the starting conditions to the spring bloom?

This is also a very good point that we are now addressing in the revised version of the manuscript. Since the cruise took place after the summer bloom in this part of the Arctic, we collected the water just below the nitracline in order to have sufficient nutrients at the beginning of the incubation to support a bloom. The nutrient concentrations at the sampling depth (38 m) corresponded to the concentrations found in the upper mixed layer in Baffin Bay in spring before the seasonal bloom (Tremblay et al. 2002, 2006). The taxonomic composition of the bloom was also similar to the one taking place in spring in this area, with *Chaetoceros* species dominating the assemblage (Von Quillfeldt, 2000). We are thus confident that the bloom that took place in our bags is comparable to the 'natural' spring bloom. But we agree that this point should have been made clearer in the paper.

L 132 – 134, new sentence: "Since the cruise took place after the summer bloom in this part of the Arctic, we collected the water just below the nitracline in order to have sufficient nutrients to support a bloom during our incubation".

L 506 – 509, new sentence: "As the initial concentrations of nutrients measured in our incubation bags were similar to the concentrations found in upper mixed layer waters in early spring before the seasonal bloom in Baffin Bay, we are confident that the bloom that took place in our bags is comparable to the natural spring bloom taking place in these waters (Tremblay et al., 2002, 2006). Furthermore, the dominance of diatoms (...)"

3. Reference to Richier et al. (2014) (Phytoplankton responses and associated carbon cycling during shipboard carbonate chemistry manipulation experiments conducted around Northwest European shelf seas) is lacking and should be included in the discussions. The work of Richier et al. is the most similar to this study in terms of the experimental techniques used. The authors do cite Hopkins & Archer (2014) which was part of the same study, but only in a DMS(P) context. The shipboard incubations of Richier et al. and Hopkins & Archer also need to be addressed in the context of this study in terms of the phytoplankton response.

We are now directly referring to the paper by Richier et al. (2014) in the revised version of the manuscript (see the following responses to the specific comments L 79 – 81 and L 575 – 581).

Specific comments and suggestions

Title: it would be more accurate to say 'DMS concentrations', as 'production' implies that the work include rate measurements.

As suggested, the title has been changed for "Impact of ocean acidification on Arctic phytoplankton blooms and dimethylsulfide concentrations under simulated ice-free and under-ice conditions".

L45 – 49: These two sentences are somewhat ambiguous and need further explanation. Why is climate change 'faster and more important' in the Arctic? In what respect?

L 50 – 53: The sentence has been modified as follows: "Due to various feedback processes, the air temperature in the Arctic above 64°N has warmed by 1.9°C between 1981 and 2012, a rate three times higher than the global average (ACIA, 2005, Ford et al., 2015). This phenomenon is known as the Arctic amplification (Cohen et al., 2014)."

L50 – 52: this sentence seems detached and slightly out of context. I see what the authors intend by it. Perhaps they could re-phrase so it says something like: 'Given that the reduction in extent and thickness of sea ice cover and the acidification of surface waters can potentially impact primary productivity, it is important to consider the associated effects on the production of biogenic climate-active gases. . .' or similar, just to change the emphasis slightly, and provide an impetus for the work.

L56 – 59: Thank you for the suggestion. The sentence has been changed for: "Given that the reduction in extent and thickness of sea ice cover and the acidification of surface waters can potentially impact primary productivity, it is important to consider the associated effects on the production of biogenic climate-active gases such as dimethysulfide (DMS) in the Arctic".

L70 – 75, and throughout: the authors make no mention of Richier et al (2014), a recent and relevant paper that should be cited.

L 79 – 81: Reference to Richier et al. has been added as follow: "(...) but negative impacts of decreasing pH on phytoplankton growth have also been reported and attributed to pH-induced alterations in algal cells physiology, acid-base chemistry, trace metal availability, ion transport, protein functions, and nutrient uptake (Doney et al., 2009; Gao and Campbell, 2014; Richier et al., 2014; Mackay et al., 2015; Thoisen et al., 2015)."

L80: re-word. Suggest: 'Emissions of DMS thus can. . .'L83: add 'atmosphere' at end of sentence (so reads 'summer Arctic atmosphere').

L 88: As suggested "The DMS emission" has been changed for "Emissions of DMS". L 91: "such as the summer Arctic" has been changed for "such as the summer Arctic atmosphere".

L97: Not necessary to cite Webb at this point as it is not a review paper. Fine to just cite the references as you specifically mention them later in the paragraph.

L 104: "Several studies have already highlighted the sensitivity of DMS production to decreases in seawater pH (Webb et al., 2015 and references therein)" has been changed for "Several studies have already highlighted the sensitivity of DMS production to decreases in seawater pH."

L99: although Archer et al. is mentioned later in the paragraph in an Arctic specific context, it would be appropriate to add it to the listed references here.

L 105 – 108: "The majority of these experimental studies revealed a negative impact of decreasing pH on DMS production (Hopkins et al., 2010; Avgoustidi et al., 2012; Webb et al., 2016)" has been changed for "The majority of these experimental studies revealed a negative impact of decreasing pH on DMS production (Hopkins et al., 2010; Avgoustidi et al., 2012; Archer et al., 2013; Webb et al., 2016)"

L142: 'submitted' would be better substituted for 'exposed'.

L156: "the phytoplankton communities were submitted to a pH gradient and two light regimes" has been changed for "the phytoplankton communities were exposed to a pH gradient and two light regimes."

L397: should read 'species'.

L 413: "specie" has been changed to "species"

L403: to improve readability, re-phrase: 'The sole exception was the LL control mesocosm. . .'

L 419 – 421: "The sole exception was the control microcosm ($pH_T$ of 8.1) exposed to LL conditions" has been changed to "The sole exception was the LL control microcosm ($pH_T$ of 8.1)".

L452: Rather than staying 'high pHT', it would be useful to state the range of pH over which the

response was observed.

L 467 – 469: "the mean $DMSP_T$ concentration decreased with increasing proton concentration, but only under the HL treatment at high $pH_T$" has been changed to "the mean $DMSP_T$ concentration decreased with increasing proton concentration but only under the HL treatment between $pH_T$ 8.1 – 7.6".

L513 – 517: this long sentence needs some re-wording as it is currently hard to follow and the English is poor in places.

L 532 – 536:

Old sentence: "During our experiment, the sharp increase in $DMSP_T$ and DMS coincided with the exhaustion of $NO_3^-$ in most of the microcosms (Fig. 3a, b). Despite $NO_3^-$ concentration still ranged between 0.9 and 2.6 $\mu$mol $L^{-1}$ at day T6 in the microscosms at lowest pH (i.e. 7.4 and 7.2), the observed peak in $DMSP_T$ concentration in those bags was of lower magnitude, reaching approximately 30 nmol $L^{-1}$ at T6 (in contrast with other microcosms, where $DMSP_T$ reached values around 100 nmol $L^{-1}$ that day)".

New sentence: "During our experiment, the sharp increase in $DMSP_T$ and DMS coincided with the exhaustion of $NO_3^-$, with the exception of the microcosms at $pH_T$ 7.4 and 7.2 under both light regimes (Fig. 3a, b). At those pH, the increase of $DMSP_T$ between T5 and T6 was of lower magnitude compared to the other microcosms and $NO_3^-$ concentrations between 0.9 and 2.6 $\mu$mol $L^{-1}$ were still measured in the bags at T6 (Fig. 7a, b)."

L524: should read 'switched'.

L 543: "switch" has been changed to "switched".

L527 – 528: needs re-wording. Suggest: 'These results also suggest that diatoms could have more difficulty in efficiently taking up/assimilating. . .'

L 546 – 547: "These results also suggest that diatoms could have more difficulty to efficiently take up/ assimilate $NO_3^-$ at lower pH" has been change to "These results also suggest that diatoms could have more difficulty in efficiently taking up/ assimilating $NO_3^-$ at lower pH".

Section 4.2: some discussion of the results in comparison to the findings of Richier et al. would be useful, as the two studies use very similar techniques – yet yield quite contrasting responses.

The following sentences were added in section 4.2:

L 575 - 581: "In contrast to our study, Richier et al. (2014) reported a negative impact of ocean acidification not only on nanophytoplankton but on picophytoplankton as well during a microcosm experiment using a similar methodology. In this study conducted with water from the northwest European shelf, lowering the pH resulted in a decrease in the abundance (cell number) and biomass (Chl *a*) of phytoplankton < 10 $\mu$m. These contrasting results could reflect differences in the initial picophytoplankton community composition and possible species-specific physiological response to OA. By contrast, (...)"

L608 – 610: Archer et al (2013) and Hopkins and Archer (2014) report rate measurements – so this statement is not correct, and their findings should be included in the discussion.

L 640: The sentence "Unfortunately, these hypotheses remain to be confirmed, given the lack of DMS production and degradation rate measurements." was deleted.

L 642 - 646: Was added to the discussion: "Results from the few previous studies where gross rate measurements were performed show no consistent effect of a decrease in pH on neither DMSP synthesis nor DMS consumption (Archer et al. 2013, Hopkins and Archer 2014). Despite the lack of rate measurements in our study, the dominance of diatoms, an algal group lacking DMSP lyase enzymes, suggests that bacteria may have played a critical role in the observed DMS dynamics."

---

## Author Comment (AC2) · 2 Mar 2017

The manuscript provides a good account of the potential effects on OA of Baffin Bay seawater in the Arctic Ocean and it's affect on various variables such as Chl a, pH, nutrients, DMSPt and DMS etc., The manuscript is well presented and figures and tables are very clearly produced. Significant changes have been highlighted in the 10 day incubation experiment. Whilst the authors state that the rapid change in pH investigated over 10 days is not representative of the gradual OA that is taking place their study does reflect potential extreme responses. However, some further acknowledgement of this should be made in the discussion and in particular acknowledge that organisms do adapt to changes which may well affect the validity of some the discussion and conclusions.

We wish to thank the referee for his positive remarks.

In response to the comment regarding the capacity of the organisms to acclimate to changes in pH, we added the following sentence:

L 628 – 634, new sentence: "However, it is important to keep in mind that our short-term experiment precludes any acclimation of the algae to their new environment, something that is likely to take place in nature with a more gradual change in pH.  In that regard, two studies have highlighted the acclimation capacity/evolutionary adaptation of the strong DMS(P) producer *Emiliana Huxleyi* to decreases in pH (Lohbeck et al., 2012; 2014). More studies are needed to fully assess how the acclimation capacity of phytoplankton will combine with short-term physiological responses to environmental stressors to shape future DMS emissions and climate."

L 730 – 731, the following passage was added in the conclusion: "...although our results do not account for the acclimation/evolutionary adaptation potential of natural microbial communities."

The abstract should contain more of the important findings mentioned in the text. Go through and highlight these changes in discussion and make sure they are included in the abstract.

This is a good point. The following sentences were added to the abstract:

- L 32 – 35: "During our experiment, a sharp increase in $DMSP_T$ and DMS concentrations coincided with the exhaustion of $NO_3^-$ in most of the microcosms, suggesting that the nutrient stress stimulated DMS(P) synthesis by the diatom community."

- L 36 – 37: "The pH-induced decreases in Chl *a* concentration suggest a decrease in net carbon fixation by diatoms under low pH conditions."

The introduction is well stated although there should be some attempt perhaps in the discussion to state why different authors find different affects of OA on phytoplankton response.

L 575 - 581, in part 4.2 "Phytoplankton community and nutrient uptake response to the pH gradient", a new sentence is discussing the contrasting results found by our study and the very similar study of Richier et al. We give there some hypotheses that could explain the contrasting responses of phytoplankton to OA observed between these studies. These hypotheses could also be valid for other OA experiments that reported contrasting results (Thoisen et al., 2015; Villafane et al., 2015)

L 575 - 581: "In contrast to our study, Richier et al. (2014) reported a negative impact of ocean acidification not only on nanophytoplankton but on picophytoplankton as well during a microcosm experiment using a similar methodology. In this study conducted with water from the northwest European shelf, lowering the pH resulted in a decrease in the abundance (cell number) and biomass (Chl *a*) of phytoplankton $< 10~\mu$m. These contrasting results could reflect differences in the initial picophytoplankton community composition and possible species-specific physiological response to OA. By contrast, (...)"

Methods. Are the expts 9 days or 10 days-it is not clear. As the authors removed the large grazers could microzooplankton affected the results? Why was alkalinity kept constant? Surely in the natural environment and in particular a bloom event alkalinity would change as well as the concentration and ion activities of some of the constituents measured?

The whole experiment lasted 10 days, but the incubation lasted 9 days: T0 was the day when we filled the bags and did the initial acidification. We then started sampling the bags at T1 for the incubation experiment. This inconsistency was corrected in the revised version of the manuscript. To avoid confusion, we are now stating that the experiment lasted 9 days.

The point about removing the large grazers is valid, although the practice is common for this type of experiment. Unfortunately, we can only speculate about how removing the large grazers may have affected our results. Nonetheless, the absence of a relationship between the abundance of heterotrophic protists and the $H^+$ and DMS concentrations suggest that grazing by microzooplankton did not have a significant influence on the observed results. For sake of clarity, a sentence was added so the readers are made aware that removing the large grazers may have affected the relative importance of microzooplankton grazing on small autotrophic and heterotrophic cells (see L 667 – 671, later in the document).

The alkalinity in our samples was kept constant only during the initial process of acidification, and we did not control alkalinity during the following 9 days. However the alkalinity varied only slightly (< 2% variation) between day 1 and day 9 in all bags. The variations could be attributed mostly to biological phosphate and silicate uptake or $CaCO_3$ precipitation/dissolution (Richier et al., 2014), $CaCO_3$ reactions being the main process responsible for non-conservative TA variation (Cross et al., 2013). However, calcareous species are believed to be absent from our experiment (see Poulin et al. 2011), so we suggest that this process would have been minimal and negligible.

We also note that the majority of the salts in seawater are non-reactive ions (see Riebesell et al., 2010), and their ion activities would have remained essentially constant during the incubation experiments, since salinity did not vary. Changes in minor/trace seawater constituent concentrations (e.g., nitrate, phosphate, silicate, iron, organic ligands, which would be biologically taken up or produced) could surely affect trace element complexation in solution, but these would not affect the pH or total alkalinity within the precision of our measurements.

Results: See the sticky notes added to the manuscript and please attend to them. Can you say what species were mainly reflected in the nanoplankton. Were any calcareous?

At the peak of the bloom, the nanoplankton were mostly composed (74% of total cells) of small (< 20 $\mu$m) centric diatoms *Chaetoceros* species. Unfortunately, we could not identify the *Chaetoceros* to the species level. We found no calcareous species in the samples, but since we preserved the samples with acidic Lugol, calcareous species may not have been preserved. Irrespective, no calcareous species have yet been identified in these waters (see Poulin et al. 2011).

Discussion and Conclusion: see the sticky notes. These parts need to be carefully gone over and some sentences modified.

The suggested modifications and re-wording have been carefully applied. Here are more specific responses to some of your comments:

- L 105: Have any of these studies considered speciation differences in CO2 in seawater? Have any of these studies measured or calculated CO2 throughout the experiment and compared with the speciation of CO2 in controls?

The concentrations of the individual species of the carbon dioxide system in solution cannot be measured directly (Dickson et al., 2007), but are derived from pH and total alkalinity measurements using the MS Excel macro $CO_2SYS$ (Pierrot et al., 2006).

None of the cited studies reported the speciation between the different species in water ($CO_2$, $HCO_3^-$ and $CO_3^{2-}$). As these studies were focused mostly on the biological impact of OA on the microbial and planktonic communities and their influence on DMS dynamics, the authors may had chosen to focus their sampling and analysis efforts on the biological community.

We calculated the speciation of $CO_2$ for all microcosms at T1, T4 and T9. As expected, $HCO_3^-$ represented around 94% of the carbonate species in all microcosms at T1, T4 and T9. The

proportion between free $CO_2$ and $CO_3^{2-}$ varied between 7- 0.6% depending of the pH, with the lowest $CO_3^{2-}$ / highest $CO_2$ values observed at the lowest pH, which is concordant with the information found in the literature (see the diagram of $CO_2$, $HCO_3^-$ and $CO_3^{2-}$ concentrations in function of pH for example). Despite these natural changes due to pH modification in the microcosms, theses percentages ($HCO_3^-$/ $CO_3^{2-}$/ $CO_2$) did not change during the incubation and remain relatively constant between T1, T4 and T9. Moreover, measuring the impact of each individual species on the monitored biological processes would have required many buffered experiments. This would have been logistically very difficult to conduct on board of the Amundsen.

- L 131: You state above seawater temperature is -1.35oC why difference?

Since our deck-incubator was cooled with circulating surface water, we had no control over the temperature. However, all HL and LL bags were in the same incubator and were therefore held at the same temperature.

L 142 – 145, new sentence added: "Since our deck incubator was cooled with circulating surface water, we had no control over the incubation temperature (mean temperature of 4.3 ± 1.6°C over the 9-day experiment). However, all bags were in the same incubator and, therefore, held to the same temperature."

- L150-152: Can you be more specific?

When $CO_2$ dissolves in the ocean, it combines with water ($H_2O$) to form carbonic acid ($H_2CO_3$). This acid then dissociates to form one proton ($H^+$) ion and one bicarbonate ($HCO_3^-$), and thus, the absorption of $CO_2$ does not impact alkalinity (e.g., Riebesell et al., 2010).

- L160-163: What about changes in ion activities and complexation effects that also affect pH? Did you test for these effects? As alkalinity was kept constant this is not what would happen in the natural environment? Could discuss these issues in discussion

Please, see our response to your general comment about the method at the beginning of this document.

- L 203-204: How much seawater was sampled for analysis and how did this loss of volume affect the results?

To minimise potential low-volume effects, we removed at most half the initial volume of each bag over the course of the 9-day experiment (ca.5 out of 10 L). Since large grazers had been removed prior to the incubation, we believe that the gradual decrease in volume had no impact on the relative abundance of the protists assemblage.

L 217: "At least half of the initial volume of the microcosms was still..." has been changed to "At least half of the initial volume of the microcosms (5L) remained ..."

- L262: How much seawater was taken?

20 mL samples were taken directly from each bag with a syringe connected to the luer-lock port of each bag.

L 276: "Water for FRRF measurements was taken" has been changed to "For FRRF measurements, 20 mL of seawater were taken (...)".

- L330: Did you calculate concentrations of H+ ions or activities?

Proton (H$^+$) concentrations were measured spectrophotometrically on the total proton concentration scale under the constant ionic medium convention (e.g. Dickson et al., 2007; section SOP 6b)

- L 338: But could it not also increase due to decreasing volume of seawater in the bags as you take more sample out for analysis? Also could plankton growth affect Ca and Mg ions? which could affect activities of H+? In natural ocean these complexation affects would be diluted but perhaps not in incubation experiments?

Plankton growth will affect macronutrient concentrations (nitrate, phosphate, silicate), but it is the uptake of $CO_2$ for organic carbon production that will affect pH most. Calcium and magnesium are major constituents of seawater and their concentrations in seawater are not significantly (< 0.5%) affected by plankton growth, except in areas of massive biogenic $CaCO_3$ precipitation such as on the Bahamas Bank and Persian Gulf. Given the presumed absence of calcareous species and the low concentration of these micronutrients in seawater, their uptake should not affect pH values within the precision our measurements. However, due to the limitation of our experiment, we could neither substantiate nor reject the possible impact of these ions complexation on the results.

- L 389: This statement seems at variance with the above?

The 1-2 days lag between the peak in chlorophyll $a$ and the peak in nanoplankton abundance did not affect much the overall correlation between the two variables. The lag probably reflects a decrease in chlorophyll $a$ synthesis (and thus chlorophyll $a$ cell quota), as the dividing cells were becoming nitrogen limited.

L 397 – 398, new sentence added: "The lag probably reflects a decrease in chlorophyll $a$ synthesis (and thus chlorophyll $a$ cell quota) as the dividing cells were becoming nitrogen limited."

- L 542: Don't you mean Si concentrations? If not how did you measure Si consumption?

L 561: "Si(OH)$_4$: NO$_3^-$ consumption" has been changed to "Si(OH)$_4$: NO$_3^-$ concentration".

- L 565: Did you measure any production of free CO2 in your expts? If not can you calculate it for the length of your expt? Is it significant?

We did not measure $CO_2$ production *per se* during the incubation. If there was any $CO_2$ production (this includes all bacterial respiration processes), this would have been implicitly captured by the change in pH and DIC over time. However, our measurements only show the net effect of $CO_2$ production and consumption. To unravel any individual processes, we would have need labelled incubations.

- L 591: But this assumes DMS is mainly produced from DMSPt? Could it not be produced directly into seawater as a stress response?

Both algae and bacteria can produce DMS from DMSP. In algae, DMS can be produced as a stress response. This stress response is, however, only present in species with DMSP lyase capacity such as the prymnesiophytes *Phaeocystis spp.* and *Emiliana hyxleyi*. So far, DMSP lyases were never found in diatoms, the dominant group during our experiment. In the open ocean, bacteria are though to be responsible for most of the transformation of dissolved DMSP (released by the algae in the medium) into DMS. For this reason, we believe that most of the DMS produced during our incubation resulted from bacterial activity. Since non-diatom species present in the assemblage may have contributed to this production, our statement refers to both direct exudation of DMS by algae as well as bacterial DMSP cleavage. To make this point clear, the sentence was modified as follow:

L 617 – 619: "Together, these results suggest that ongoing OA will have a stronger impact on the algal and bacterial DMSP transformation into DMS than on the synthesis of DMSP by algae (...)"

- L 602: This is a long bow from a 10 days experiment. It suggests that organisms don't adapt to change.

L 628: Deleted passage: "...during the next centuries, with OA potentially counteracting the predicted stimulation of DMS production due to sea-ice retreat and the consequent increase in primary production (Six et al., 2013)."

Passage added to the discussion: please, see our response to your general comment at the beginning of the document.

- L 611: You can get a measure of DMSPt and DMS production rates or consumption rates by taking the concentration measurements on the different days to see if production or consumption varied much as a function of the elapsed time between measurements. You are comparing concentrations of DMSPt /DMS with changes in ion activities of H+ not an ideal comparison. As DMSPd did not change much it seems unlikely that bacterial activity had much effect being swamped by the increases in Chl a?

Since both DMSP and DMS are produced and consumed by distinct pathways, day-to-day changes in the size of their pools only allow a calculation of net changes in DMSP and DMS, not their gross changes.

Because pH is simply equal to the negative logarithm of $[H^+]$, the linear relationship between different dependant variables and pH, as often indicated in the literature, is not appropriate. This

is the reason why we choose to compare DMSPt and DMS with $H^+$ concentration ($[H^+]$). A similar type of representation was also adopted by Archer et al. (2013) and Hopkins and Archer (2014) during their mesocosm/microcosm experiments.

- L 617: Some reference should be made to changes in these ratios also observed in polar waters by Jones et al. (1998)

L 654 – 656, new sentence added: "Although the decrease of this ratio could also be due to an increased grazing of diatoms by microzooplankton (Jones et al., 1998), we found no significant relationship between the micrograzers and $H^+$ or DMS."

- L 631: Whilst you removed large grazers you presumably still had microzooplankton present?

This is right. The abundance of heterotrophic protists (mostly ciliates and choanoflagellates) was measured at T0, T5 and T9 in 6 pH treatments during the study (8.07, 7.61 and 7.22 at both high and low light). Their abundance varied from 786,008 to 15,254,481 cells $L^{-1}$. However, we found no significant relationship between their abundance and $H^+$ or DMS concentrations.

To clarify our argument, some details were added to the discussion:

L 667 – 671: "Heterotrophic protists were present in the microcosms, with abundances varying between ca. 786,008 to 15,254,481 cells $L^{-1}$ (data not shown). Although removing large grazers before the incubation may have affected the relative importance of the microzooplankton grazing on phytoplankton, no relationship between protists abundance and $H^+$ or DMS could be found."

- L 638: a large biomass of centric diatoms could have increased DMS and affected DMSPt. Why not use the % of each species found and apply this % to DMS and DMSPt production (concentrations measured) and see if any trends.?

The large diatom bloom that took place during our experiment certainly contributed to the increase in DMSPt, but since DMSP lyases have never yet been identified in diatoms, they are likely not directly responsible for the associated increases in DMS concentrations.

- L 639: Jones et al (1998) suggest an inverse correlation occurs in polar waters of the Southern Ocean for diatoms and dinoflagellates. Could something like this occur in your incubation bags?

We cannot respond to this question since the few dinoflagellates present in the water at the beginning of the experiment quickly disappeared. The dinoflagellates seem to be too fragile to survive the initial filling of the bags.

- L 656: Yes this is an important point and makes sense in a melting sea ice environment (see Vance et al. 2013?)

L 697 – 699, new sentence added: "These results suggest that phytoplankton exiting the ice pack would not necessarily experience a light shock as previously noted by others (Vance et al., 2013, Galindo et al., 2016)."

- L 680: Too much speculation. This last sentence should be removed. As Chl a increased and is a function of SRD.

L 719: The following sentence was deleted: "These results further suggest that SRD may not be the main factor driving net DMS production in Arctic waters, similar to results from the northeast Atlantic, where Belviso and Caniaux (2009) found that the SRD accounted for only 19% to 24% of DMS variations during the summer."

- L 682: very rapid OA!

L 721: "During this study, we demonstrated that OA could negatively impact" has been changed to "During this study, we demonstrated that a rapid decrease in surface water pH could negatively impact"

- L 688: but Chl *a* and DMSPt increased as Chaetocerous increased?

It is true that Chl *a* and DMSPt concentrations increased in parallel with the bloom of *Chaetoceros spp.* during the experiment, but the amplitude of the peak in Chl *a* and DMSPt decreased as the pH decreased.

- L 712: This last bit seems to be at variance with what you have stated earlier?

We agree with the reviewer.

L 750 – 753, new sentence added: "(...), our results show that Arctic diatoms may bloom under light conditions much lower than the one tested here. This apparent capacity of Arctic diatoms to growth under extremely low light conditions should be explored in future studies."

Overall I would recommend publication with attention paid to the minor comments. Also the authors should end their discussion with what future studies should concentrate on wrt. Baffin Bay to extend the field and make these expts more relevant to actual conditions in the field.

L 753 - 755, new sentence added: "As short-term impacts of OA on the DMS cycle become clearer, future studies should focus on assessing the potential adaptation and tolerance mechanisms of microbial DMS(P) producers, mechanisms that likely develop on a time scale closer to the natural OA rate."

Please also note the supplement to this comment: http://www.biogeosciences-discuss.net/bg-2016-501/bg-2016-501-RC2- supplement.pdf